# Sortilin limits EGFR signaling by promoting its internalization in lung cancer

Hussein Al-Akhrass[1], Thomas Naves [1], François Vincent[1,2], Amandine Magnaudeix [1], Karine Durand[1,3], François Bertin[4], Boris Melloni[2], Marie-Odile Jauberteau[1,5] & Fabrice Lalloué[1]

Tyrosine kinase receptors such as the epidermal growth factor receptor (EGFR) transduce information from the microenvironment into the cell and activate homeostatic signaling pathways. Internalization and degradation of EGFR after ligand binding limits the intensity of proliferative signaling, thereby helping to maintain cell integrity. In cancer cells, deregulation of EGFR trafficking has a variety of effects on tumor progression. Here we report that sortilin is a key regulator of EGFR internalization. Loss of sortilin in tumor cells promoted cell proliferation by sustaining EGFR signaling at the cell surface, ultimately accelerating tumor growth. In lung cancer patients, sortilin expression decreased with increased pathologic grade, and expression of sortilin was strongly correlated with survival, especially in patients with high EGFR expression. Sortilin is therefore a regulator of EGFR intracellular trafficking that promotes receptor internalization and limits signaling, which in turn impacts tumor growth.

[1] EA3842 Homéostasie Cellulaire et Pathologies and Chaire de Pneumologie Expérimentale, Université de Limoges, Faculté de Médecine, 2 Rue du Dr. Raymond Marcland, 87025 Limoges CEDEX, France. [2] Service de Pathologie Respiratoire, Centre Hospitalier et Universitaire de Limoges, 87042 Limoges CEDEX, France. [3] Service d'Anatomie Pathologique, Centre Hospitalier et Universitaire de Limoges, 87042 Limoges CEDEX, France. [4] Service de Chirurgie Thoracique et Cardio-vasculaire, Centre Hospitalier et Universitaire de Limoges, 87042 Limoges CEDEX, France. [5] Service d'Immunologie, Centre Hospitalier et Universitaire de Limoges, 87042 Limoges CEDEX, France. Hussein Al-Akhrass and Thomas Naves contributed equally to this work. Marie-Odile Jauberteau and Fabrice Lalloué jointly supervised this work. Correspondence and requests for materials should be addressed to T.N. (email: thomas.naves@unilim.fr)

Aberrant activation of tyrosine kinase receptors (TKRs), which mediate signal transduction between cells and their microenvironment, occurs in 76% of all cases of lung adenocarcinomas[1]. TKRs relay the extracellular cues into the cell, leading to regulation of intracellular processes related to cell proliferation, migration, and survival[2]. The epidermal growth factor receptor (EGFR) is the archetypal TKR[3, 4]. EGFR signaling is triggered by binding of its growth factor ligands, such as epidermal growth factor (EGF), leading to the autophosphorylation of tyrosine residues in its cytoplasmic tail and thereby inducing cell signaling. Subsequently, EGFR is internalized[5], and both the endocytic route and the fate of EGFR are regulated by adaptor proteins that dock with the tyrosine kinase domain[6].

The rapid internalization and degradation of the EGFR are under tight spatiotemporal control to limit cell proliferation promoted by mitogen activated protein kinases (MAPKs)[7–9]. This negative feedback mechanism, governed by ligand-induced lysosomal degradation of EGFR, ensures signal termination and counteracts the oncogenic and transforming role of EGFR[10–12]. Accordingly, high-EGFR expression is a common feature of multiple cancers. Furthermore, inactivation of sorting proteins, which regulate both the duration and the intensity of EGFR signaling, plays a causal role in EGFR-induced promotion of tumor growth by sustaining proliferative signaling, a hallmark of cancer[13–18].

Because multiple facets of EGFR trafficking remain unresolved[19], and EGFR internalization represents a crucial step for signal termination, we investigated the role of sortilin[20–22] in EGFR regulation following EGF-induced EGFR internalization. Sortilin, a member of the vacuolar protein sorting 10 (VPS10) protein family of sorting receptors[23], shuttles between the plasma membrane and the trans-Golgi network (TGN)[21, 22, 24]. The VPS10 domain constitutes the entire luminal domain of sortilin[25], which is considered to be a multifaceted sorting receptor involved in neurotrophin TKR trafficking in neurons[26]. In a previous report, we showed that sortilin also facilitates both the transport and loading of EGFR into extracellular vesicles containing exosome specific markers[27]. Because EGFR is not present in exosomes derived from sortilin-depleted cells, we focused on the function of sortilin in EGFR intracellular trafficking. Our results reveal that sortilin regulates EGFR by controlling its internalization from the plasma membrane, thereby limiting proliferative signaling, an essential driving force behind tumor aggressiveness. Moreover, we found that low expression of sortilin is associated with more aggressive lung adenocarcinoma tumors. Hence, sortilin expression represents a favorable prognostic marker in lung adenocarcinoma patients.

## Results

**EGF stimulation promotes EGFR and sortilin interaction.** Sortilin has been implicated in several protein sorting pathways between the plasma membrane, endosomes, and the TGN[28]. Based on findings from an earlier report in which we observed that sortilin participates in loading of EGFR into exosomes[27], and because exosome synthesis depends on endosome trafficking[29], we speculated that sortilin is involved in sorting a pool of EGFR that increases upon ligand-induced EGFR internalization. To achieve complete EGFR endocytosis and avoid endosome arrest and EGFR recycling via EGFR-inhibited autophagy[30], we stimulated A549 human non-small cell lung carcinoma cells with EGF under normal serum conditions, analyzed the canonical EGF-induced pathways of active EGFR in whole-cell lysate (WCL), and investigated whether EGF stimulation promoted the interaction between EGFR and sortilin. As expected, EGFR

activation induced MAP kinase signaling, as evidenced by elevated ERK1/2 phosphorylation downstream of EGFR activation (Fig. 1a, WCL panel). Furthermore, EGF stimulation promoted EGFR internalization, as reflected by the reduction in EGFR levels following lysosomal degradation[31].

Consistent with the initial hypothesis, immunoprecipitation (IP) performed on the same lysates confirmed that EGFR co-immunoprecipitated with sortilin under basal conditions, and that this interaction was strengthened following EGF stimulation (Fig. 1a, IP panel). In addition, we used a proximity ligation assay (PLA) to confirm the EGFR—sortilin interaction over a time course (Fig. 1b, insets 1–1 to 5–2). In the images, the red spots indicate sites of proximity ligation amplification, reflecting the interaction between EGFR and sortilin, which was significantly strengthened after just 2 min of EGF stimulation (Fig. 1c), suggesting that the EGFR—sortilin interaction occurs in the early stage of EGFR internalization. These results were further confirmed by fluorescence resonance energy transfer (FRET) assay (Supplementary Fig. 1a, b). To determine where EGFR and sortilin interact, we first investigated the co-localization of EGFR and sortilin co-localization with various organelle markers, including the early endosome (EE) marker Rab5 (Ras-related protein Rab5), the late endosome/lysosome marker LAMP2 (Lysosome-associated membrane glycoprotein 2), and the TGN marker TGN46, in the presence or absence of EGF (Fig. 1d). Under basal conditions, EGFR is mainly localized at the cell membrane (Fig. 1d, insets 1–3). As expected, upon EGF stimulation, internalized EGFR overlapped primarily with the Rab5 and LAMP2 markers (Fig. 1d, insets 7, 8), indicating that the majority of active EGFR resided in the endosomal and lysosomal degradation pathways, consistent with the initial results (Fig. 1a). We also observed weak co-localization of EGFR with TGN46 (Fig. 1d, inset 9) consistent with its translocation through the Golgi[16]. However, to determine the subcellular localization of sortilin, we transfected A549 cells with a sortilin-GFP fusion protein, thereby avoiding cross-reactivity with the secondary antibody (Fig. 1d, insets 4–6 and 10–12). Under basal conditions, sortilin-GFP overlapped primarily with TGN46 and Rab5, in agreement with published reports that less than 10% of the total sortilin pool is expressed at the plasma membrane before being recycled back to the TGN through endosomes[22, 24, 32]. Upon EGF stimulation, sortilin-GFP overlapped continuously with TGN46, and sortilin-GFP significantly co-localized with Rab5-positive compartments (Fig. 1d, insets 10, 12) rather than with the lysosomal marker LAMP2 (Fig. 1d, inset 11). To rule out the possibility that EGF stimulation induces EGFR and sortilin interaction in the TGN after post-translational modifications, we used the protein synthesis inhibitor cycloheximide (CHX). PLA performed on EGF-stimulated cells in the presence of CHX revealed no signal variation (Supplementary Fig. 1c–e). These results were also supported by the absence of variation in mRNA and protein levels of either EGFR or sortilin after EGF stimulation (Supplementary Fig. 1f). Hence, we excluded the possibility of an interaction between EGFR and sortilin in the TGN, where the majority of the sortilin resides. These observations were also supported by co-localization analysis based on Mander's overlap coefficients (Fig. 1e) and the co-localization of EGFR, Rab5, and sortilin in sortilin-GFP-transfected cells following EGF stimulation (Fig. 1f, insets 1–1 and 1–2). Post-nuclear supernatants from A549 cells stimulated or not with EGF were separated onto a 0–30% Iodixanol gradient and subjected to ultracentrifugation (Supplementary Fig. 1g). In the absence of EGF stimulation, intracellular EGFR was distributed diffusely between the EE and the TGN enriched fractions, corresponding to EGFR turnover, whereas sortilin was mainly in the TGN fraction. Interestingly, upon EGF stimulation, EGFR and sortilin

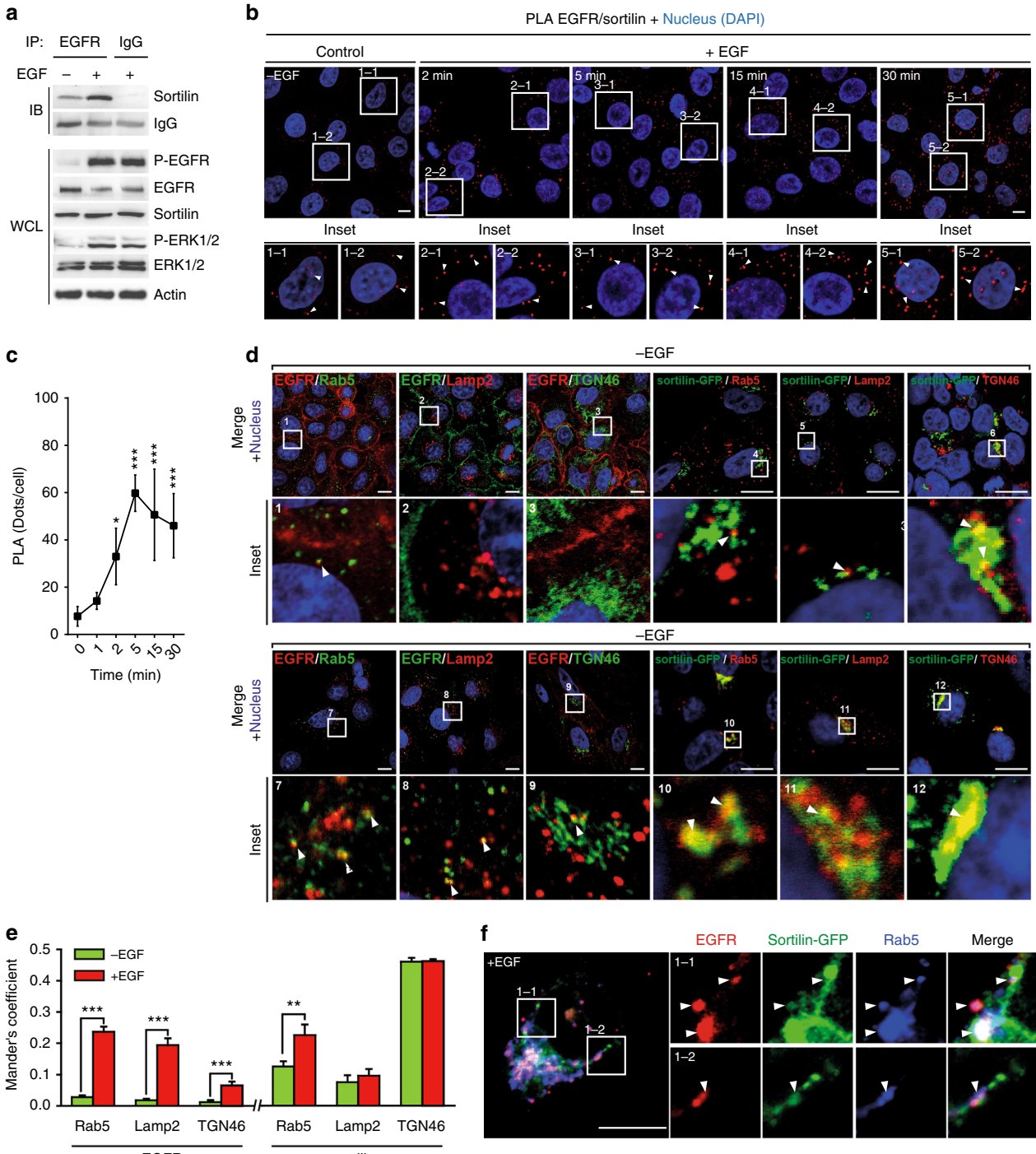

**Fig. 1** EGF promotes the EGFR—sortilin interaction. **a** A549 cells grown in complete cell culture media were stimulated or not with EGF (50 ng/mL) for 30 min. Immunoprecipitations (IP) were performed using anti-EGFR antibody, and the immunocomplexes were immunoblotted (IB) using anti-sortilin antibody (top). In parallel, immunoblots for P-EGFR, EGFR, sortilin, P-ERK, and ERK were performed on whole-cell lysates (WCL); the isotypic lane Immunoglobulin G (IgG) represents the IP control. **b** Proximity ligation assays (PLA) were performed on A549 cells, non-stimulated or stimulated with EGF (50 ng/mL) for 2, 5, 15, and 30 min. Red spots indicate sites of proximity ligation assay amplification, reflecting the EGFR—sortilin interaction (white arrows). Scale bar, 10 μm. **c** Quantification of PLA time course, in comparison with non-stimulated cells. **d** A549 cells were stimulated or not with EGF (50 ng/mL) for 30 min, and then co-immunolabeled for EGFR and markers of the early endosome (Rab5), the late endosome/lysosome (LAMP2) and the trans-Golgi network (TGN46). For sortilin labeling, A549 cells were transiently transfected with sortilin-GFP to adapt a set of functional antibodies, and then immunolabeled for the same markers. Scale bar, 5 μm. **e** Quantitative analysis of EGFR or sortilin-GFP co-localization with the aforementioned organelle-specific markers. **f** A549 cells were transiently transfected with sortilin-GFP, and then stimulated with EGF (50 ng/mL) for 30 min. Next, cells were fixed and co-immunolabeled for EGFR and Rab5. Scale bar, 10 μm. All values represent means ± SD, Student's t-test *P < 0.05; ***P < 0.001. Each experiment has been repeated at least three times

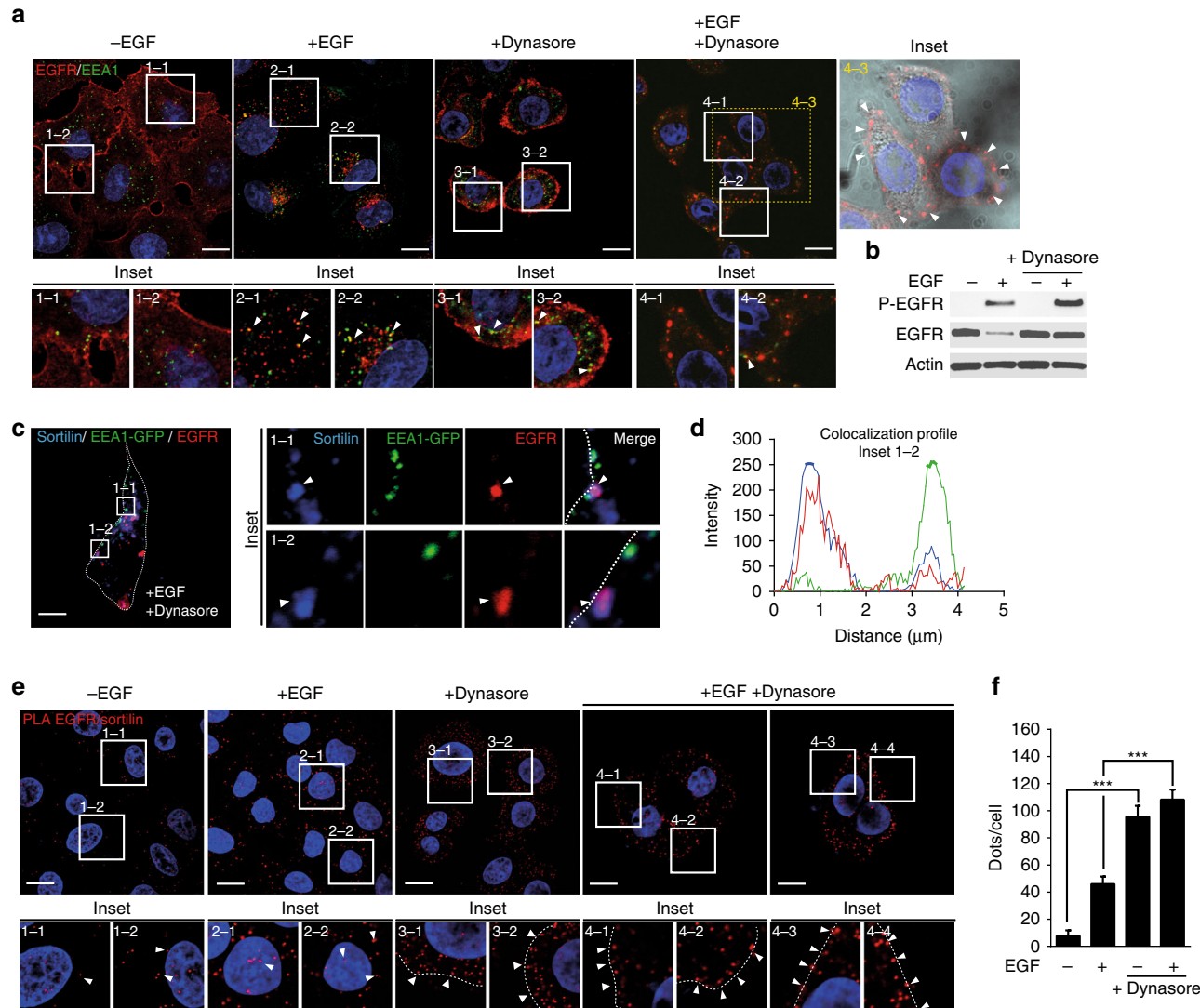

**Fig. 2** EGFR interacts with sortilin at the cell surface. **a** A549 cells were pretreated or not with the cell-permeable dynamin inhibitor Dynasore (40 μM) for 2 h, and then stimulated or not with EGF (50 ng/mL) for 15 min. Cells were immunolabeled for EGFR and the early endosome marker EEA1, and then analyzed by confocal microscopy. Scale bar, 10 μm. Inset 4–3: Bright-field image of A549 cells; white arrows show EGFR clusters at the cell surface. Scale bar, 10 μm. **b** A549 cells were pretreated or not with Dynasore (40 μM) for 2 h, and then stimulated or not with EGF (50 ng/mL) for 15 min. The cell lysates were analyzed by western blotting for P-EGFR and EGFR. **c** A549 cells were transfected with EEA1-GFP, pretreated with Dynasore (40 μM) for 2 h, then stimulated with EGF (50 ng/mL) for 30 min. Next, cells were co-immunolabeled for EGFR and sortilin. Scale bar, 10 μm. The co-localization profile is shown in **d**. **e** Proximity ligation assays (PLA) were performed on A549 cells under the same conditions described above in **a**. Scale bar, 10 μm. **f** PLA quantification in comparison with A549 non-stimulated cells. All values represent means ± SD, Student's t-test ***P < 0.001. Each experiment has been repeated at least three times

are distributed into two populations according to EGFR engagement in intracellular trafficking and especially in the EE.

Together, these results suggest that EGF strongly induces the EGFR—sortilin interaction following EGFR activation and internalization, and that sortilin interacts with EGFR during the early stage of EGFR internalization.

**EGFR interacts with sortilin at the plasma membrane.** To determine whether the early stage of EGFR endocytosis promotes the EGFR—sortilin interaction, we inhibited endocytosis using Dynasore, a cell-permeable dynamin inhibitor[33]. Indeed, inhibition of dynamin prevents clathrin-mediated EGFR endocytosis[34] and endosome maturation in the early stage of endocytosis[35]. Thus, Dynasore inhibits both endosome maturation and endocytosis of newly formed endosomes. In non-stimulated or Dynasore-pretreated cells, EGFR was mainly localized at the cell surface (Fig. 2a, insets 1–1, 1–2, 3–1, and 3–2). EGF stimulation elicited co-localization between EGFR and the early endosome antigen 1 (EEA1) (Fig. 2a, insets 2–1 and 2–2), whereas dynamin inhibition impaired EGF-induced EGFR endocytosis, as confirmed by the presence of EGFR clusters at the cell surface and a reduction in the overlap between the EGFR and EEA1 signals (Fig. 2a; insets 4–1, 4–2, and 4–3 (bright-field)). As expected, following plasma membrane retention of the EGFR upon Dynasore treatment, EGF-stimulated cells exhibited both sustained EGFR phosphorylation and reduced EGFR degradation (Fig. 2b). Furthermore, Dynasore alone did not induce EGFR phosphorylation (Fig. 2b). In A549 cells, we observed an overlap between EGFR and sortilin that co-localized poorly with EEA1 (EEA1-GFP), as revealed by confocal microscopy (Fig. 2c, insets 1–1 and 1–2) and the co-localization profile (Fig. 2d). Together,

these results suggest that EGFR interacts with sortilin in forming endosomes upstream of EEA1 recruitment.

Therefore, we next performed PLA assays to investigate whether the EGFR—sortilin interactions were maintained by Dynasore treatment. Interestingly, Dynasore alone induced a robust EGFR—sortilin interaction that is significantly stronger than the basal interaction in control cells or that induced by a pulse of EGF (Fig. 2e, insets 1–1 to 3–2 and Fig. 2f). Furthermore,

this interaction was still independent of EGFR activation, as observed in the results described above, and EGFR was not phosphorylated in the presence of Dynasore alone (Fig. 2b). Surprisingly, addition of EGF to Dynasore–pretreated cells did not alter the EGFR—sortilin interaction in comparison to that in cells treated with Dynasore alone (Fig. 2e, insets 4–1 to 4–4 and Fig. 2f). Because sortilin cycles continually between the plasma membrane and TGN[22, 24, 32], Dynasore might impair the normal

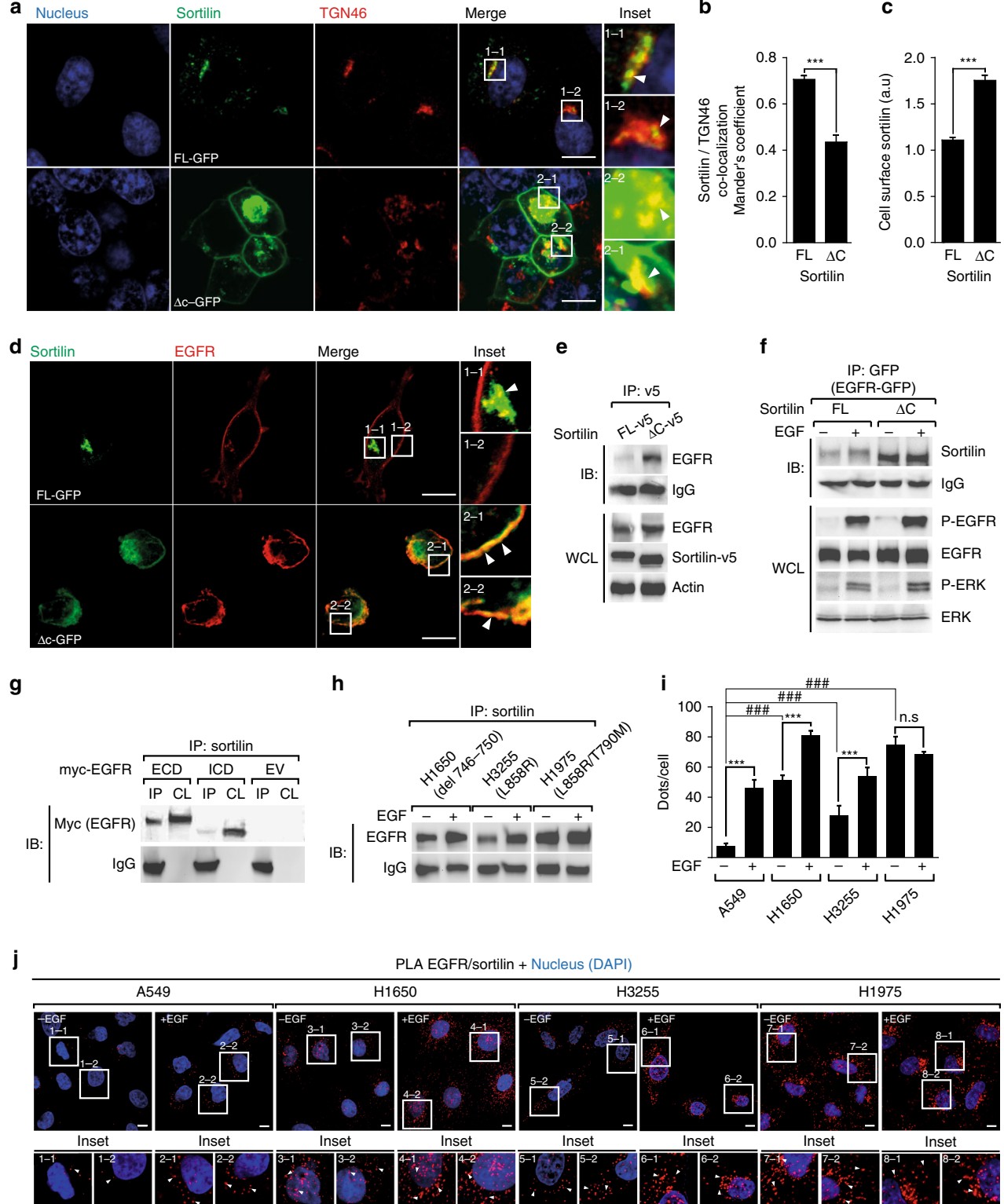

trafficking of EGFR—sortilin through their endocytic route, forcing them to maintain their interaction. These results suggest that sortilin participates in the internalization of unliganded EGFR, which is 10-fold slower than that of EGF-stimulated receptor[36].

Together, these results indicate that in A549 cells, the EGFR—sortilin complex takes place at the plasma membrane. However, Dynasore could alter endosome maturation, TGN or other trafficking intermediates driving to accumulated structures close to the plasma membrane. Hence, to better delineate the location of EGFR—sortilin, we used a version of sortilin that cannot be endocytosed, leading to accumulation of sortilin at the cell surface.

**Cell surface-enriched sortilin strongly interacts with EGFR.** To investigate the mechanism underlying the EGFR—sortilin interaction, we used the non-tumor cell line HEK293, which expresses low levels of both endogenous EGFR and sortilin[37, 38]. Because sortilin cycles from the plasma membrane to the TGN, and in light of our previous observation that Dynasore alone unveiled the EGFR—sortilin interaction, we overexpressed a sortilin mutant lacking the intracellular C-terminal domain (Δc-sortilin) but containing the extracellular VPS10 sorting domain[23]. Because the C-terminal domain of sortilin controls its internalization, Δc-sortilin cannot be recycled from the plasma membrane[24, 39]. We transfected HEK293 cells with either full-length (FL) or truncated (Δc) GFP-tagged sortilin, and then monitored co-localization of both proteins with TGN46. As expected, FL-sortilin was largely localized to the perinuclear region, along with TGN46, whereas Δc-sortilin exhibited plasma membrane retention, as indicated by Mander's overlap coefficient (Fig. 3a, insets 1–1 to 2–2, Fig. 3b, c). Next, we overexpressed both EGFR and sortilin to evaluate the interactions between EGFR and FL- or Δc-sortilin. Interestingly, EGFR immunostaining overlapped with GFP-tagged Δc-sortilin signal at the cell surface (Fig. 3d, insets 1–1 to 2–2). Next, we transiently transfected cells with EGFR and FL- or a Δc-v5 tagged sortilin. Without EGF stimulation, IP revealed a stronger interaction between EGFR and the Δc-sortilin that accumulated at the cell surface (Fig. 3e). We then generated HEK293 stable cell lines that overexpressed both GFP-tagged EGFR and untagged FL- or Δc-sortilin. In both stable cell lines, the canonical pathway of EGF-induced EGFR activity was unaltered, as attested by the proper phosphorylation of MAPK ERK1/2 (Fig. 3f, WCL panel). Interestingly, IP confirmed the EGFR/Δc-sortilin interaction, with an increase in the EGFR/FL-sortilin interaction occurring exclusively after EGF stimulation (Fig. 3f, IB panel), as previously observed in A549 cells.

Taken together, these data demonstrate that the EGFR—sortilin interaction occurs at the cell surface, and is strengthened when sortilin is enriched at the cell membrane regardless of EGF stimulation.

**Sortilin and EGFR interact by their extracellular domains.** To further delineate the domain of interaction between EGFR and sortilin, we transiently co-transfected Δc-sortilin with either the extracellular (ECD) or intracellular domain (ICD) of EGFR in HEK293 cells. Interestingly, EGFR-ECD mostly immunoprecipitated with sortilin (Fig. 3g), indicating that this interaction involves the ECD of EGFR, and the sortilin VPS10 sorting domain, which constitutes the entire ECD of sortilin, is implicated in several intracellular sorting processes[22, 23, 40]. These results suggest that point mutations or deletion of the EGFR intracellular tyrosine kinase domain, frequently reported in lung adenocarcinoma, cannot prevent the EGFR—sortilin interaction. Indeed, mutant EGFR proteins are strongly tyrosine-phosphorylated even in the absence of ligand[41]. Consistent with the constitutive hyperphosphorylated state of EGFR, sortilin and EGFR interacted in H1650 (delE746-A750), H3255 (L858R), and H1975 (L858R/T790M), as attested by immunoprecipitations and PLA (Fig. 3h, j). Interestingly, in the absence of ligand stimulation, the EGFR—sortilin interaction remained stronger in EGFR-mutated cell lines (Fig. 3i) than in A549 cells, suggesting that the greater ability of the EGFR mutants to be internalized[42, 43] facilitates their interaction with sortilin. Surprisingly, although EGF stimulation significantly promoted the EGFR—sortilin interaction in H1650, H3255, and A549 cells, it had no such effect in H1975 cells (Fig. 3i).

Together, these results suggest that the VPS10 sorting domain of sortilin interacts with EGFR, and that sortilin acts as a sorting receptor for EGFR following its internalization.

**Sortilin depletion retains EGFR at the cell surface.** Several lines of evidence indicate that sortilin is involved in EGFR trafficking, independently of EGF-induced EGFR internalization. Consequently, the absence of sortilin could limit EGFR endocytosis from the plasma membrane. Interestingly, in sortilin-depleted A549 cells[27], EGFR internalization was perturbed, as revealed by immunostaining showing EGFR clusters at the plasma membrane irrespective of EGF stimulation (Fig. 4a, insets 1–1 to 2–2). Confocal images showed that sortilin depletion significantly altered the ratio of whole-cell EGFR vs. intracytoplasmic EGFR (Fig. 4b). These observations were supported by fluorescence-activated cell sorting (FACS) analysis in which we measured the cell surface level of EGFR to monitor its

**Fig. 3** C-terminally truncated sortilin strongly interacts with EGFR at the cell surface independently of ligand stimulation. **a** HEK293 cells were transiently transfected with either full-length (FL) or C-terminally truncated (Δc) sortilin-GFP. Cells were fixed and immunolabeled for the trans-Golgi network marker TGN46, and then analyzed by confocal microscope. Scale bar, 10 μm. **b** Bars show the Mander's coefficient, indicating that FL-sortilin-GFP co-localized with TGN46 to a greater degree than Δc-sortilin-GFP. **c** Bars show quantification of cell surface GFP intensity. **d** Strong interaction between EGFR and Δc-sortilin at the plasma membrane. HEK293 cells were transiently co-transfected with EGFR and FL- or Δc sortilin-GFP, and then fixed and immunolabeled for EGFR; immunofluorescence was analyzed by confocal microscopy. Scale bar, 10 μm. **e** HEK293 cells were transiently co-transfected with EGFR and FL or Δc sortilin-v5. Immunoprecipitations (IP) were performed using anti-v5 antibody, and immunocomplexes were analyzed by western blotting using anti-EGFR antibody. In parallel, immunoblots of EGFR and sortilin-v5 were performed on whole-cell lysate (WCL). **f** HEK293 cells were transiently co-transfected with EGFR-GFP and FL- or Δc sortilin. Next, cells were stimulated or not with EGF (50 ng/mL) for 30 min and immunoprecipitated using anti-GFP antibody. Immunocomplexes were analyzed by western blotting for sortilin. In parallel, immunoblots of P-EGFR, EGFR, P-ERK, and ERK were performed on WCL. **g** HEK293 cells were transiently co-transfected with ΔC sortilin and the myc-tagged extra- or intracellular domain (ECD and ICD, respectively). Next, cell lysates (CL) were immunoprecipitated (IP) with anti-sortilin and immunoblotted with anti-myc. **h** NSCLC cell lines harboring EGFR deletion (H1650) or point mutations (H3255, H1975) were stimulated or not with EGF (50 ng/mL) for 30 min. Cell lysates were immunoprecipitated (IP) using anti-sortilin, and immunocomplexes were analyzed by western blotting with anti-EGFR. **i** PLA quantification performed on NSCLC cell lines stimulated or not with EGF (50 ng/mL) for 30 min. **j** Proximity ligation assays (PLA) were performed on NSCLC cells under the same conditions as described in **h**. All values represent means ± SD, Student's t-test ***P < 0.001. Each experiment has been repeated at least three times

internalization following EGF stimulation over a 60-min time course (Fig. 4c). Our results revealed that EGFR internalization was significantly slower in sortilin-depleted cells than in controls (Fig. 4c, point 30 and 60 min). Accordingly, endocytic assay using fluorescent EGF supports the delay of

EGFR internalization in sortilin-depleted cells (Supplementary Fig. 2a).

Subsequently, we investigated whether sortilin depletion would impair the EGFR canonical signaling pathway. Interestingly, under basal conditions, sortilin-depleted cells exhibited sustained

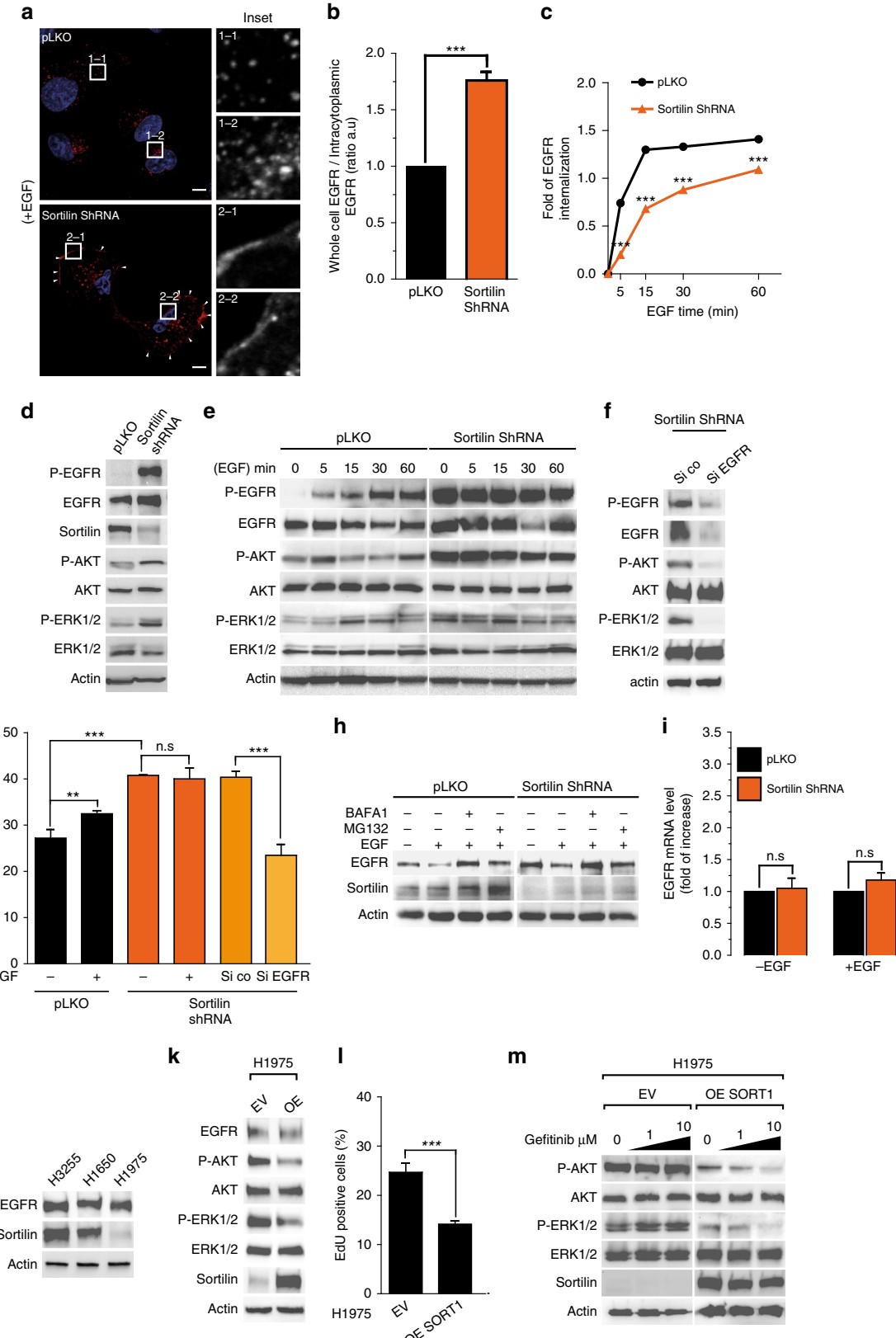

EGFR signaling, as evidenced by the hyperphosphorylated state of EGFR and MAP kinases AKT and ERK (Fig. 4d). Following EGF stimulation, control cells exhibited an increase of EGFR phosphorylation, followed by MAPK activation, as evidenced by elevated phosphorylation of both AKT and ERK1/2 (Fig. 4e). Furthermore, whole-cell levels of EGFR decreased with increasing treatment time, consistent with the lysosomal degradation of the protein. Importantly, disruption of EGFR internalization delayed EGFR turnover and increased both the activity and amount of EGFR. Strikingly, sortilin depletion maintained EGFR in a constitutive hyperphosphorylated state, in agreement with its plasma membrane retention[14] (Fig. 4d, e). Hence, EGFR-induced constitutive signaling, as attested by phosphorylation of the MAP kinases AKT and ERK1/2 (Fig. 4d, e), suggesting that EGFR plasma membrane retention drives proliferative signaling in the absence of sortilin. As expected, silencing of EGFR in A549 sortilin-depleted cells decreased MAP kinase signaling (Fig. 4f). Accordingly, sortilin depletion significantly increased 5-ethynyl-2′-deoxyuridine (EdU) incorporation, reflecting a higher rate of cell proliferation, in comparison with control cells (Fig. 4g). However, EGF stimulation did not increase cell proliferation of sortilin-depleted cells, consistent with the basal hyper-activated status of both EGFR and its downstream network (Fig. 4g). Interestingly, EGFR silencing significantly decreased the proliferation of sortilin-depleted cells, supporting the idea that the higher proliferation rate of these cells is governed by sustained EGFR proliferative signaling. We reproduced these results in two other EGFR-amplified cell lines, the human glioblastoma U87-MG and the epidermoid squamous A431 cells (Supplementary Fig. 2b–g).

However, to rule out the possibility that sortilin depletion impairs the EGFR degradative pathway, we individually inhibited the lysosome and proteasome. Bafilomycin A1, a lysosomal acidification inhibitor, decreased EGFR degradation upon EGF stimulation in both control and sortilin-depleted cells; by contrast, the proteasome inhibitor MG132 had no such effect. Conversely, sortilin degradation was limited upon proteasome inhibition (Fig. 4h). These results suggest that sortilin is transiently associated with, and acts as a sorting receptor from the plasma membrane, and that the EGFR degradative pathway remains functional despite the absence of sortilin. Furthermore, sortilin depletion did not increase the EGFR mRNA level irrespective of the presence of EGF (Fig. 4i), supporting the idea that EGFR accumulation in sortilin-depleted cells corresponds to

mis-sorting of EGFR from the plasma membrane. Our results suggested that sortilin limits EGFR proliferative signaling. Interestingly, H1975 cells expressed a lower level of endogenous sortilin than other EGFR-mutated cell lines (Fig. 4j). Because H1975 harbors the EGFR T790M point mutation, which is insensitive to EGFR tyrosine kinase inhibitor (TKI) therapy, we transiently overexpressed sortilin in these cells. Transient sortilin overexpression in H1975 cells decreased both AKT and ERK phosphorylation under basal conditions (Fig. 4k), as well as cell proliferation (Fig. 4l), suggesting that sortilin alters the malignant behavior of these cells. Thus, despite the strong EGFR—sortilin interaction in the H1975 cell line (Fig. 3i), the weak expression of endogenous sortilin seems insufficient to counteract the EGFR proliferative signaling pathway. Strikingly, sortilin overexpression reversed the gefitinib-resistance phenotype of H1975 cells, as evidenced by reduced AKT and ERK phosphorylations in comparison with control cells (Fig. 4m).

Together, these results demonstrate that loss of sortilin impairs EGFR internalization and sustains its proliferative signaling.

**Sortilin depletion accelerates tumor growth in vivo.** Previous studies show that sortilin is required for EGFR turnover in vitro, and that sortilin downregulation sustains EGFR proliferative signaling. To determine whether the loss of sortilin is important for tumor maintenance in vivo, we examined the effect of sortilin depletion on human adenocarcinoma xenografts. For this purpose, we subcutaneously engrafted A549 control or sortilin-depleted cells into nude immunodeficient mice, and then followed tumor growth. As expected, sortilin-depleted cells exhibited much faster tumor growth than control cells from the fifteenth day onward (Fig. 5a, b). Consistent with this, tumors derived from sortilin-depleted cells were significantly larger on day 25 (Fig. 5a, b). To further assess the proportion of proliferative cells in each group of engrafted mice, we performed immunohistochemical (IHC) staining for Ki-67, a marker of proliferation. In agreement with their faster growth, sortilin-depleted cells had an elevated Ki-67 proliferative index (~80%) relative to control cells (~25%) (Fig. 5c, d). Together, these results show that loss of sortilin dramatically increases cell proliferation, and consequently tumor growth, in vivo.

**Sortilin downregulation is associated with poorer prognosis.** Our results obtained on animal models reveal that loss of sortilin

---

**Fig. 4** Loss of sortilin greatly perturbs EGFR internalization and dramatically promotes EGFR signaling. **a** A549 cells expressing shRNA targeting sortilin and control cells (pLKO) were stimulated with EGF (50 ng/mL) for 30 min, and then fixed and immunolabeled for EGFR. Immunofluorescence was analyzed by confocal microscopy. The ratio between whole-cell and intracytoplasmic EGFR intensities, reflecting EGFR membrane retention, is quantified in **b**. **c** Sortilin-depleted and control A549 cells (pLKO) were stimulated with EGF (50 ng/mL) over a 60 min time course. At each time point, cell surface EGFR was stained at 4 °C using Alexa Fluor 488 anti-EGFR, and then analyzed by flow cytometry. Curves represent mean fluorescence intensity in sortilin-depleted cells relative to the control. **d** Cell lysates from sortilin-depleted (sortilin shRNA) or control A549 cells (pLKO) were immunoblotted with the indicated antibodies. **e** A549 sortilin-depleted and control cells (pLKO) were stimulated with EGF (50 ng/mL) over a 60 min time course. Cell lysates were analyzed by western blotting for components of the canonical EGFR signaling pathway using the indicated antibodies. **f** Sortilin-depleted cells were transfected with control siRNA (Si co) or EGFR siRNA. Cell lysates were analyzed by western blotting with the indicated antibodies. **g** Representative histograms of cell proliferation, as determined by EdU incorporation. Sortilin-depleted cells transfected or not with EGFR siRNA and control A549 cells (pLKO) were stimulated with EGF (50 ng/mL) for 1 h, and then fixed and treated for EdU incorporation. Percentages of EdU-positive cells were calculated by flow-cytometric analysis. **h** Sortilin-depleted (sortilin shRNA) or control A549 cells (pLKO) were pretreated with bafilomycin A1 (BAFA1) or MG132 for 2 h, and then stimulated or not with EGF (50 ng/mL) for 30 min. Cell lysates were analyzed by western blotting for EGFR and sortilin protein expression. **i** Quantitative PCR analysis of EGFR expression in sortilin-depleted and control A549 cells (pLKO) with or without EGF stimulation (50 ng/mL for 30 min). Results are presented in terms of fold change after normalization against *HPRT* mRNA. **j** Cell lysates from H3255, H1650, and H1975 were analyzed for EGFR and sortilin protein expression. **k** H1975 cells were transfected or not with *SORT1* overexpression vector, and cell lysates were then analyzed by western blotting for the indicated proteins. **l** Representative histograms of cell proliferation, as determined by EdU incorporation in sortilin-overexpressing and control H1975 cells. **m** Sortilin-overexpressing and control H1975 cells were treated with increasing doses of gefitinib for 24 h, and cell lysates were analyzed by western blotting for the indicated proteins. All values represent means ± SD, Student's *t*-test ***P < 0.001. Each experiment has been repeated at least three times

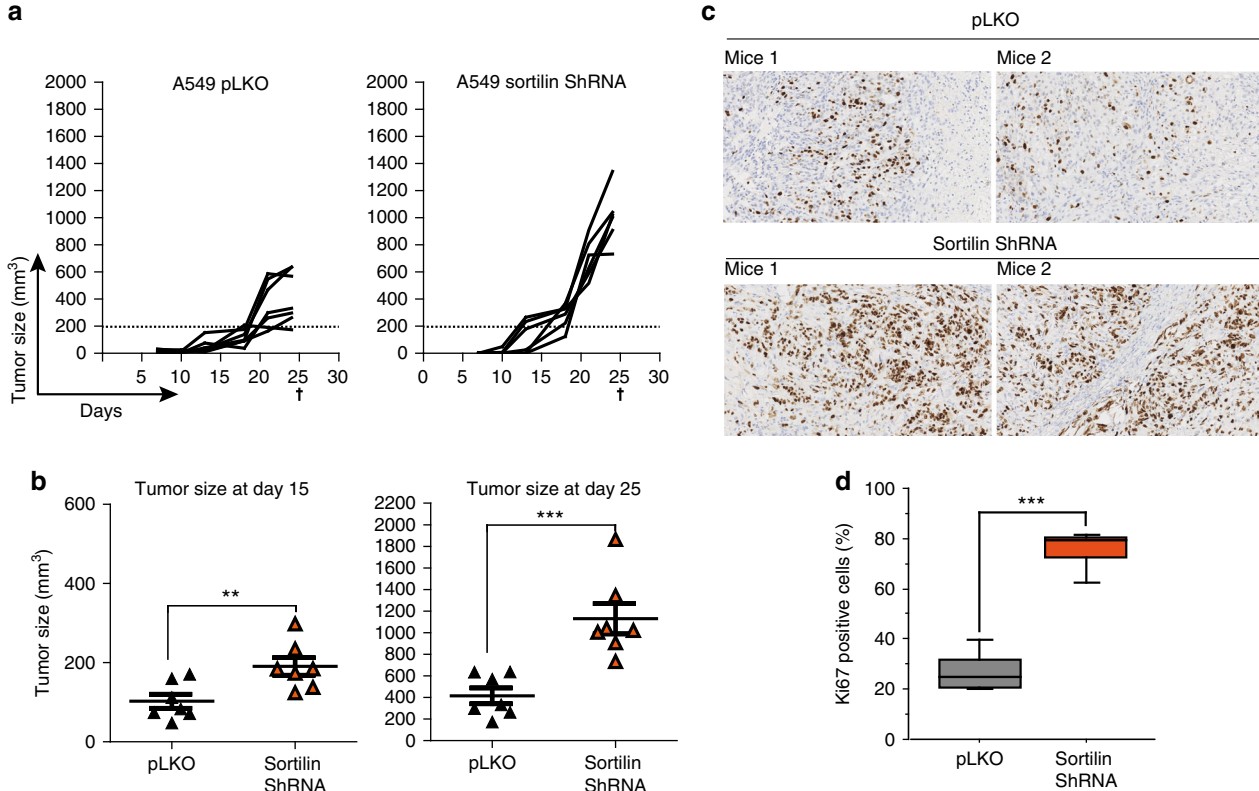

**Fig. 5** Sortilin depletion dramatically accelerates tumor growth in vivo. **a** Tumor growth curves showing evolution of the tumor volume after engraftment of $1 \times 10^6$ sortilin-depleted or control A549 cells (pLKO). **b** Grouped column scatter shows the mean difference tumor volume between sortilin-depleted and control cells, 15 and 25 days after engraftment. **c** 25 days after engraftment, all mice were killed, and tumors were excised and subjected to immunohistochemical analysis. Tumors from sortilin-depleted or control A549 cells were stained using anti-Ki-67 antibody. Percentages of Ki-67-positive cells are quantified in **d**. All values represent means ± SD, Student's t-test **$P < 0.01$, ***$P < 0.001$. Each experiment has been repeated at least three times

is associated with elevated cellular proliferation and accelerated tumor growth. To determine whether this phenomenon is also relevant to human cancer, we performed IHC analysis on 78 patients with non-small cell lung adenocarcinomas (NSCLCs). We analyzed sortilin expression according to grade: grades I–III correspond to well-differentiated, moderately differentiated, and poorly differentiated pathology, respectively; this grade increases with the tumor aggressiveness. We complemented this approach by calculating the percentage of Ki-67-positive nuclei. Interestingly, sortilin expression decreased significantly with pathologic grade (Fig 6a, b, insets 1–3), whereas the percentage of positive nuclei was significantly higher in grade III than in grades I and II (Fig. 6c, d insets 4–6). These results support the relationship between grade and aberrant cell proliferation, which is associated with poor prognosis and survival in patients with lung adeno-carcinomas[44]. These observations were further supported by the pattern of TTF-1 expression. TTF-1 is highly expressed in lung adenocarcinomas, but loss of TTF-1 is associated with tumor aggressiveness and reduced median survival[45, 46] (Supplementary Fig. 3: as with sortilin, expression of TTF-1 decreases with grade). Thus, reduced expression of sortilin in NSCLC patients was correlated with increased cellular proliferation, accelerated tumor growth, and TTF-1–negative poorly differentiated tumors. Collectively, the phenotypes of sortilin-downregulated tumors were associated with an NSCLC subtype with poorer prognosis.

In light of these results, we performed statistical analysis of publicly available adenocarcinoma data sets from the MSKCC cBioPortal database[47, 48], based on The Cancer Genome Atlas[1] (TCGA) and the gene expression omnibus (GEO). We manually sorted datasets based on overall survival and expression of

SORT1, the gene encoding sortilin. The Kaplan–Meier method was used to plot the overall survival of adenocarcinoma patients ($n = 673$) stratified by expression of SORT1, and statistical significance was assessed by log-rank test[49, 50]. High expression of sortilin (black curve) was significantly correlated (log-rank $p = 3 \times 10^{-9}$) with better survival, in contrast to low expression (orange curve) (Fig. 6e). From these results, we analyzed publicly available data to investigate whether sortilin expression could be affected by oncogenic drivers such as KRAS or EGFR amplification, as observed in lung adenocarcinoma. Although our statistical tests supported no significance of KRAS mutations, they highlighted an increase in SORT1 expression in EGFR-overexpressing tumors (Fig. 6f). Next, we investigated SORT1 expression in a cohort of patients with low or high-EGFR expression. Our analyses revealed that sortilin expression was significantly higher in patients with EGFR amplification (Fig. 6g). Because EGFR amplification is correlated with poor outcome, we sorted and curated patients ($n = 522$) with high-EGFR expression ($n = 30$). Strikingly, Kaplan–Meier analysis revealed that high-SORT1 expression in EGFR-overexpressing patients significantly increased overall survival (Fig. 6h). Because 75% of tumors with EGFR amplification harbor EGFR mutations[51, 52], we analyzed the TCGA data to investigate sortilin expression in these patients. Interestingly, lung adenocarcinomas with EGFR TKI—sensitive mutations expressed high levels of SORT1 (Fig. 6i).

Given that EGFR is often associated with initiation and the progression of NSCLC[53, 54] and sortilin attenuates EGFR proliferative signaling, as observed above in a TKI-resistant cell line (Fig. 4k, i, m), sortilin expression in tumors potentially represents a useful predictive marker of patient outcome.

## Discussion

In this study, we identified sortilin as a key regulator of EGFR trafficking and demonstrated that it limits EGFR proliferative signaling. Our finding supports a model in which sortilin sorts EGFR, modulating its accumulation at the cell surface, and thereby prevents autocrine and sustained signaling, both of which are hallmarks of cancer. The role of sortilin in plasma membrane EGFR regulation and the putative underlying mechanism are summarized in Fig. 7. In this model, ligand binding accelerates routing of EGFR by sortilin toward rapid internalization and

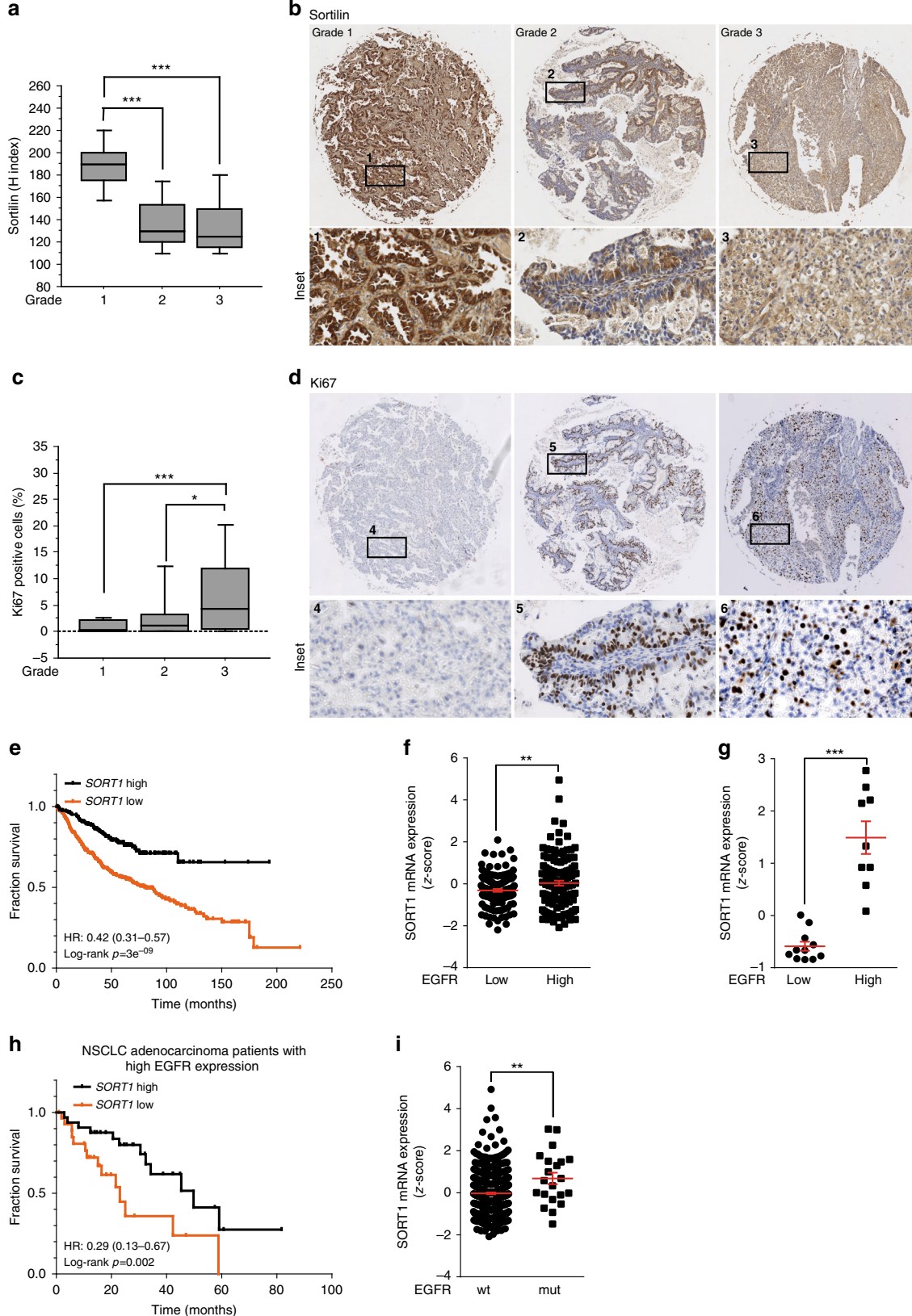

degradation. This inhibitory mechanism is of central importance to the control of signal duration and intensity. Several mechanisms can result in activation of TKR pathways during initiation and progression of NSCLC, including TKR amplification, upregulated autocrine signaling, and mutations[36, 40, 42]. Both histopathological and molecular genetics studies have shown that aberrant activation of TKRs is more frequent in lung adenocarcinomas than in other cancers[1, 55]. EGFR plays key roles in processes related to tumorigenesis, including cell proliferation, cell survival, and metastasis. Because the EGFR downstream signaling network is one of the most frequently deregulated pathways in human cancer, and because multiple facets of EGFR trafficking remain unresolved[19], the roles played by sorting proteins such as sortilin in EGFR regulation have attracted increasing attention. At the cellular level, the EGFR—sortilin interaction is supported by the results of multilayered analysis of the EGFR interactome following ligand-induced EGFR trafficking[56].

The initial site of the EGFR—sortilin interaction remains to be elucidated. Because sortilin resides mainly in the TGN, it was tempting to speculate that sortilin regulates the anterograde transport of EGFR signaling. Our observations made by inhibiting endocytosis or in cells expressing a recycling-defective variant of sortilin indicated that the two proteins interact primarily at the plasma membrane. Interestingly, the EGFR—sortilin interaction was independent of EGF-induced EGFR phosphorylation and endocytosis and did not trigger EGFR phosphorylation. In the absence of sortilin, almost all EGFR remained at the cell surface, supporting the idea that sortilin is not required to localize EGFR to the plasma membrane following post-translational modification. Thus, these observations indicate that sortilin is not a ligand for EGFR, and is instead involved in turnover of EGFR at the plasma membrane.

Downregulation of sortilin affected EGFR internalization and increased the proportion of the receptor localized to the cell surface. Plasma membrane retention of EGFR subjects the receptor to free ligands, and sustains both its proliferative and potential autocrine signaling. Therefore, loss of sortilin promotes EGFR proliferative signaling and accelerates tumor growth in vivo. Overexpression of sortilin in TKI-resistant cells resulted in strong decreases in cell proliferation and gefitinib resistance. Sortilin in the mutant EGFR lung cancer context appears to be acting as a tumor suppressor or inhibiting malignant behavior, while in other cancers it appears to act as an oncogene or promote malignant behavior[57–59]. These differences will need to be resolved in future studies. Lung adenocarcinoma patients with weak expression of sortilin have poorly differentiated tumors with accelerated proliferation. Overall, the pathologic characteristics of sortilin-downregulated tumors correspond to a subtype of adenocarcinoma with poorer prognosis. Interestingly, in patients whose tumors express high levels of *EGFR*, which is associated with poor prognosis value, higher expression of *SORT1* significantly improved the overall survival.

In summary, our findings describe for the first time the role of sortilin in limiting proliferative signaling by EGFR, and reveal a connection between sortilin expression and patient outcome. The striking role of sortilin in EGFR trafficking should be investigated in the context of other TKRs that are involved in initiation and progression of lung adenocarcinoma.

## Methods

**Immunoblotting and immunoprecipitation.** For immunoblotting, cells were washed with ice-cold phosphate-buffered saline (PBS, Gibco, France), and then lysed in cell lysis buffer (Cell Signaling) containing 1% protease inhibitor cocktail (Roche, France) and phosphatase inhibitor (Sigma, France). Cell lysates were sonified (Vibra-Cell Sonifier, set at 50% amplitude) three times (2 s each, with at least 1 min of rest on ice pulses) and clarified by centrifugation at 18,000×g. Solubilized proteins (30 μg) were subjected to SDS-PAGE and western blot analysis using antibodies specific for sortilin (BD Bioscience, #612101, 1/500), P-EGFR (Tyr 1068, #3777, 1/1000; Cell Signaling, Ozyme, France), EGFR (Cell Signaling, #4267, 1/1000, Ozyme or Life Technologies, clone H11 #MA5-13070, 1/500, Fischer Scientific, France), pERK1/2 (Thr202/Thr204, #4370, 1/1000, Cell Signaling, Ozyme), ERK1/2 (Cell Signaling, #9102, 1/1000, Ozyme), pAKT (Ser 473, #4060 1/1000, Cell Signaling, Ozyme) AKT (Cell Signaling, #4691, 1/1000, Ozyme), EEA1 (Cell Signaling, #3288, 1/1000, Ozyme), Rab5 (Cell Signaling, #3547, 1/1000, Ozyme), LAMP2 (Santa Cruz Biotechnology, sc-18822, 1/200, Tebu, France), TGN46 (Sigma, #T7576), v5, GFP tag (Life Technologies, respectively #MA5-15253, 1/1000, and #MA5-15256, 1/1000, Fischer Scientific), and actin (Sigma, #A2066, 1/10000) (used as a loading control). Immunoreactive bands were detected with horseradish peroxidase (HRP)-conjugated secondary antibodies (Dako, 1/1000, Agilent, France) in the presence of enhanced chemiluminescence substrate.

For IP, cells were washed with ice-cold PBS and lysed in IP lysis buffer (50 mM Tris-HCl, pH 7.5, 150 mM NaCl, 1% Triton X-100, 2 mM EDTA; all reagents from Sigma) containing 1% protease inhibitor cocktail (Roche) and phosphatase inhibitor (Roche). Lysates were clarified by centrifugation at 18,000×g. Cell lysates were precleared for 1 h at 4 °C with 10% (v/v) cocktail from a non-protein A—producing *Staphylococcus aureus* strain (Sigma). Precleared lysates were clarified by centrifugation for 5 min at 18,000×g. A volume of 1 mL of Protein A Sepharose beads (Sigma) was washed three times with IP buffer and resuspended in 1 mL of IP buffer. Then, 200 μg protein lysate and 2 μg antibody were solubilized in 1 mL IP buffer and 50 μL of previously prepared Protein A Sepharose beads were added to the suspension and incubated overnight at 4 °C with gentle rocking. Control lysate was mixed with normal mouse-IgG or normal rabbit-IgG and beads. The next morning, immunoprecipitates were washed three times with 1 mL of IP lysis buffer at 4 °C, and then boiled in loading buffer (Bio-Rad) at 95 °C for 5 min. The beads were removed by centrifugation, and immunoprecipitated lysates were subjected to SDS-PAGE and western blot analysis (Bio-Rad).

**Cell culture and treatments.** Cell lines HEK293T, A549, H1650, and H1975 were obtained from American Type Culture Collection (ATCC), whereas H3255 was generously provided by Sylvie Gazzeri of the Albert Bonniot Institut (France). All cell lines were maintained in Dulbecco's modified Eagle's medium GlutaMAX (Gibco) supplemented with 10% fetal bovine serum (IDbio), 1% non-essential amino acids (Gibco), and antibiotics (IDbio). All cells were cultured in a humidified incubator set at 5% CO₂ and 37 °C. Cells were cultured under serum

---

**Fig. 6** Sortilin expression decreased with the tumor aggressiveness. **a** Boxplot diagram represents the quantification (Hirsch index) of sortilin expression in human lung adenocarcinoma (*n* = 78) and highlights the high expression of sortilin at low grade, as seen in representative images in **b**: grade I, well-differentiated; grade II, moderately differentiated; grade III, poorly differentiated. Magnification ×50, insets ×200. **c** Boxplot diagram of the percentage of Ki-67-positive nuclei, reflecting cancer cell proliferation, and tumor aggressiveness in the same patients shown in **b** with representative images **d**. Magnification ×50, insets ×200. **e** Kaplan–Meier curves of overall survival were constructed for the following groups of patients with high (black curve, *n* = 248) or low (orange curve, *n* = 425) *SORT1* expression, using the online tool at kmplot.com. **f** In silico analysis of *SORT1* expression in lung adenocarcinoma with high or low *EGFR* expression; data were obtained from the MSKCC cBioPortal for Cancer Genomics database or Genome Expression Omnibus. **g** Quantitative PCR analyses for *SORT1* and *EGFR* expression in a cohort of patients (*n* = 20) with high or low *EGFR* expression. **h** Kaplan–Meier curves showing the survival benefit provided by high-*SORT1* expression (black curve, *n* = 32) relative to low *SORT1* expression (orange curve, *n* = 30) on a subset patients with high-*EGFR* expression; data were obtained from the lung adenocarcinoma cohort in the MSKCC cBioPortal database (*n* = 522). **i** In silico analysis for *SORT1* expression in lung adenocarcinoma with or without *EGFR* TKI−sensitive mutations, using data from the MSKCC cBioPortal for Cancer Genomics database or Genome Expression Omnibus. For Kaplan–Meier curves, the log-rank (Mantel-cox) is used test to determine the statistical significance of association sortilin high or low expression and overall survival within lung adenocarcinoma patients, *P < 0.05 **P < 0.01 ***P < 0.001, Hazard Ratio (HR) is reported in time-to-event analysis. For Boxplot diagrams, median values are indicated by the transverse line within the box. Student's *t*-test *P < 0.05, **P < 0.01, ***P < 0.001. Each experiment has been repeated at least three times

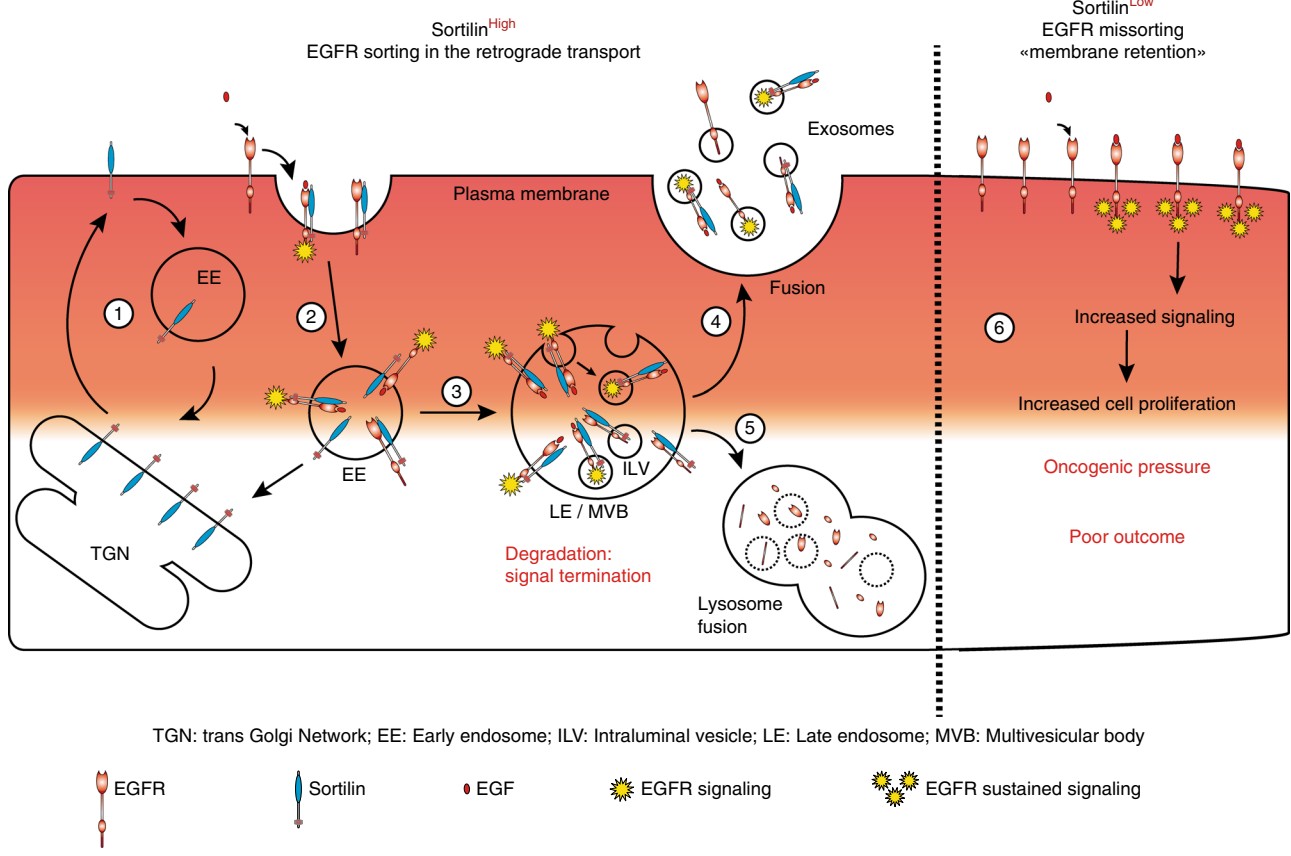

**Fig. 7** Model of sortilin function in regulation of EGFR at the cell surface Representative scheme in which sortilin acts as a key regulator of EGFR retrograde transport. Sortilin is mainly localized at the trans-Golgi network (TGN), and the plasma membrane fraction of sortilin cycles continually between the cell surface and the TGN via the endosomes (1). Sortilin binds both unstimulated and stimulated EGFR to allow their internalization (2). Thus, EGFR undergoes intracellular trafficking and sortilin mediates its loading in intraluminal vesicles (3). This ultimately results either in EGFR release through exosomes (4) or receptor degradation (5), thus ensuring signal termination. Conversely, sortilin downregulation impairs EGFR internalization (6), and EGFR consequently retained at the cell surface. Moreover, in sortilin-downregulated cells, EGFR exists in a hyperphosphorylated state, and transduces a constitutive survival signal that promotes tumor growth

conditions prior to stimulation with EGF (Life Technologies) at 50 ng/mL for 30 min. For treatments with chemical compounds, A549 cells were pretreated with 100 μM Dynasore (Sigma) for 2 h prior to EGF stimulation (50 ng/mL, 30 min).

**Cell proliferation assays and flow cytometry.** Cell proliferation was measured using the Click-it EdU Alexa Fluor 488 Flow Cytometry Assay (Invitrogen). Briefly, cells ($2 \times 10^5$ per well) were incubated overnight on six-well plates, incubated with EdU for 2 h, and then stimulated for 1 h with EGF (50 ng/mL). Proliferation was assessed by flow cytometry on a FACS Calibur instrument (BD Biosciences), and the data were analyzed using the Cell Quest software (BD Biosciences). Each experiment was repeated at least three times. Expression of cell surface EGFR was monitored by flow cytometry on living cells without permeabilization. After washing in ice-cold PBS, cells were immunolabeled on ice for 10 min with APC-conjugated anti-EGFR (Biolegend, #BLE352906, 1/100, Ozyme) in PBS containing 2% BSA. After two further washes in ice-cold PBS containing 2% BSA and one wash in ice-cold PBS, cells were suspended in PBS and analyzed by flow cytometry on a FACS Calibur instrument; data were analyzed using the Cell Quest software. Measurements were compared to the isotopic control (APC-conjugated anti-mouse IgG1, Biolegend clone MOPC-21 #BLE400122, 1/100, Ozyme) to determine background and positivity thresholds. Each experiment was repeated at least three times.

**Lysosome and proteasome inhibition.** A549 cells were cultured in a humidified incubator set at 5% $CO_2$ and 37 °C and pretreated with bafilomycin A1 (Sigma) (50 nM) or MG132 (Sigma) (10 μM) for 2 h prior to EGF stimulation (50 ng/mL, 30 min).

**Plasmids and lentivirus-mediated RNA interference.** For both transient and stable transfection, cells were transfected using the JetPei transfection reagent (Polyplus transfection, Ozyme). Lentivirus-mediated RNA interference was used to generate stable sortilin-knockdown cell lines. Briefly, HEK293T cells were

transfected with a mix containing the SORT1 ShRNA transfer plasmid (TRCN0000005295, Sigma) and the MISSION lentiviral packaging mix (Sigma) as recommended by the manufacturer and previously described[27, 60]. Cell supernatants containing lentivirus particles were collected each day during 3 days before to be concentrated in 1 mL of PBS by using the Lenti-X Concentrator reagent (Clontech, Ozyme) following the manufacturer's recommendations. Then, $5 \times 10^5$ cells were infected in complete media containing 8 μg/mL of polybrene (Sigma) and 50 μL of concentrated lentivirus during 48 h before to be selected by puromycin (1 μg/mL, Sigma). Wild-type EGFR was a gift from Matthew Meyerson[61] (Addgene plasmid #11011); EGFR-GFP was a gift from Alexander Sorkin[62] (Addgene plasmid #32751); EGFR-ECD (1–644) and EGFR-ICD (645–1186) were a gift from Mien-Chie Hung[63] (Addgene plasmid #42666 and #42667); and sortilin Full-Length and Δc constructs were gifts from Finan et al.[39].

**RNA interference.** For interference assays, sortilin-depleted A549 cells were transfected using INTERFERin (Polyplus transfection, Illkirch, France). Each transfection used 100 nM siRNA against EGFR (SignalSilence EGF Receptor siRNA I, Ozyme), or control siRNA (SignalSilence Control siRNA (Unconjugated), Ozyme).

**Quantitative reverse transcription-PCR.** The Qiagen RNeasy kit was used to isolate total RNA from cells (Qiagen). Single-stranded cDNA was prepared using the high capacity cDNA reverse transcription kit (Applied Biosystems). The reaction was stopped by incubation at 95 °C for 5 min. Approximately 100 ng of cDNA was used for each PCR reaction, performed with TaqMan (Applied Biosystems) on an ABI Step One Plus real-time thermal cycler (Applied Biosystems). PCR primers and probes for EGFR (Hs01076090_m1) and HPRT (Hs02800695_m1) were from Life Technologies, and the primer/probe set for sortilin was designed for this study.

**Immunofluorescence and confocal microscopy analysis**. Cells grown on glass coverslips and were washed twice in ice-cold PBS before fixation in methanol or 4% paraformaldehyde for 10 min on ice. After fixation, the cells were washed with wash buffer solution, PBS containing 1% (w/v) BSA, and blocked for 30 min with PBS containing 3% BSA (IDbio). The cells were immunolabeled at 4 °C overnight with the indicated primary antibody, anti-EGFR (Cell Signaling, Ozyme, #4267, 1/100), anti-sortilin (Abcam, #ab16640, 1/100, France) in blocking solution. The following morning, the cells were washed three times (PBS, 1% BSA), and primary antibodies were labeled either with Alexa Fluor 594–conjugated anti-rabbit-IgG or Alexa Fluor 488–conjugated anti-mouse-IgG antibodies (1:1000; Life Technologies) for 2 h at room temperature. The cells were washed three times (PBS, 1% BSA). Finally, the cells were mounted using Fluoroshield mounting media (Sigma) containing 4′,6-diamidino-2-phenylindole (DAPI) to stain the nuclei. For endocytic assay, biotinylated EGF, complexed to Alexa Fluor 647, was used following manufacturer's instructions (Life Technologies, France, #E35351). Fluorescence images were obtained using epifluorescence microscopes (Zeiss Axiovert) equipped with a laser-scanning confocal imaging system (Zeiss LSM 510 META or LSM800). Mander's coefficients were calculated using the Zeiss LSM 510 META or ZEN software (Zeiss) on non-saturated pictures with an optical slice of 0.8 μm. At least 30 cells were acquired for each condition. Cell surface expression of either EGFR or sortilin (calculated from the difference between the whole-cell and the intracellular means of fluorescence) were analyzed using the ImageJ software (ImageJ).

For PLA, the cells were fixed with 4% paraformaldehyde for 10 min, and then permeabilized for 30 min on ice in 0.1% Triton X-100 (Sigma) in PBS. Subsequently, the cells were washed with PBS, and blocking solution (2% BSA in PBS) was applied for 30 min at 37 °C in a humidified chamber. Primary antibodies against EGFR (mouse monoclonal, 1:100, Life Technologies) and sortilin (rabbit polyclonal, 1:100, Abcam) diluted in blocking solution were added, and the sample was incubated for 30 min at 37 °C. The cells were then washed with buffer A of the Duolink II proximity ligation assay kit (Olink Bioscience, Sigma). Subsequently, the Duolink II PLA probe anti-mouse Minus and the Duolink II PLA probe anti-rabbit Plus were added to the cells, and the sample was incubated for 60 min at 37 °C. To link the two probes, the cells were washed in buffer A and incubated for 30 min at 37 °C in Duolink II ligation buffer diluted in filtered distilled water containing ligase. Following ligation, the cells were washed in buffer A, and then incubated for 100 min at 37 °C with the Duolink II orange amplification buffer containing polymerase. The cells were then washed three times in buffer B and mounted with in-situ mounting media containing DAPI. Quantitative analyses obtained from each independent sample were performed using the ImageJ software (NIH, Bethesda) based on the mean fluorescence values. At least 50 cells were acquired for each condition, and the results are presented as ratios relative to the control cells.

**Endosome/subcellular fractionation**. Cells were washed with ice-cold PBS (pH 7.4), rapidly swelled with osmotic buffer (10 mM Tris-HCl (Sigma), pH 7.4), and scraped into homogenization buffer (10 mM Tris-HCl, pH 7.4, 1 mM EGTA, 0.5 mM, EDTA, 0.25 M sucrose; all reagents from Sigma) containing 1% protease inhibitor cocktail and phosphatase inhibitor. The disrupted cells were homogenized using a Dounce homogenizer (IKALabortechnik), and post-nuclear supernatants (PNS) were obtained by centrifugation of the homogenate for 10 min at 1000 g. PNS were centrifuged at 10,000×g for 20 min at 4 °C to pellet the plasma membrane, mitochondria, and rough endoplasmic reticulum. The PNS were then fractionated in a discontinuous 10–30% (w/v) OptiPrep (Sigma) density gradient as described by Li and Donowitz[49].

A 50% (w/v) OptiPrep working solution (WS) was prepared by mixing 5 volumes of OptiPrep with 1 volume of diluent solution (0.25 M sucrose, 6 mM EDTA, 60 mM Tris-HCl, pH 7.4). Then, the WS of 50% OptiPrep was diluted in homogenization media (HM) (0.25 M sucrose, 1 mM EDTA, 10 mM Tris-HCl, pH 7.4) to obtain nine solutions containing different percentages of OptiPrep (10%, 12.5%, 15%, 17.5%, 20%, 22.5%, 25%, 27.5%, and 30%). The OptiPrep cushions were layered manually in an ultracentrifuge tube, with higher concentrations at the bottom and lower concentrations at the top. The PNS were diluted into the HM and loaded onto the top of the light cushion before to at ultracentrifugation at 100,000×g for 16 h at 4 °C. The following morning, 20 fractions of the 10 layers were collected, and the proteins were precipitated using 20% (w/v) of trichloroacetic acid (Sigma) and 10% (w/v) of ice-cold acetone (Sigma). The precipitated proteins were pelleted by centrifugation at 18,000×g for 30 min at 4 °C, then washed twice in ice-cold acetone by repeated centrifugation of 18,000×g for 2 min at 4 °C. Protein pellets were solubilized in the loading buffer and boiled at 95 °C for 5 min. Proteins were subjected to SDS–PAGE in the order of collection (i.e., from light to the heavy subcellular membrane vesicles), and western blot analysis was performed with the corresponding antibodies.

**Mice and in vivo tumor growth**. Nude female mice were obtained from Janvier Labs (France). Mice were housed in specific pathogen–free conditions, and experiments were done in accordance with the guidelines of the French Veterinary Department. Young mice (6–8 weeks of age) were injected subcutaneously in the left thigh with $1 \times 10^6$ cells in 20 μL of PBS. Tumor volume (=Length × Width × ((Length + Width)/2)) was measured twice weekly. Mice were killed 25 days after injection. Tumors were collected, fixed with formaldehyde, embedded in paraffin, and processed for IHC.

**Patients and immunohistochemistry**. Lung adenocarcinoma tissue microarrays containing 48 cases of lung adenocarcinoma, with information about pathological grade and TTF-1 IHC results, were provided by US Biomax (BCS04017a; US Biomax, United States). In addition, 30 lung adenocarcinoma tumors in paraffin-embedded blocks were obtained from the Tumor Bank (Biolim) of Limoges University Hospital, under protocols approved by the Institutional Review Board (AC-2013-1853, DC-2011-1264) of the Anatomo-Pathology department of CHU Dupuytren Limoges. All patients were informed of the use of their tissue samples in research studies. Immunohistochemical and hematoxylin/eosin staining was performed on 5-μm-thick consecutive sections. Antibodies against sortilin (Alomone, Israël, 1/175, #ANT-009), EGFR (Cell Signaling, Ozyme, #4267, 1/25), and Ki-67 (Dako, clone MIB-1, #M7240, 1/200) were used for tissue labeling on a Leica Bond-Max using the Bond Polymer Refine Detection kit. For sortilin and EGFR labeling, samples were pretreated with ER1 for 5 min; for Ki-67 labeling, samples were pretreated for 20 min with ER2. Labeling was performed with the same device (Leica Bond-Max), and slides were mounted in a non-aqueous mounting medium. Images were acquired on a Hamamatsu slide scanner. Each image capture was visually quantified by the adapted Hirsh score method[64], performed in a triple-blind manner. To evaluate the percentage of Ki-67-positive nuclei, the publicly available software Immunoratio[65] was used.

**Bioinformatic analyses**. Overall survival and vital status were obtained using the Kaplan−Meier Plotter for lung cancer (kmplot), and computed as a function of sortilin expression using data extracted from GEO ($n = 673$ patients) data sets with accession numbers GSE14814, GSE19188, GSE29013, GSE30219, GSE31210, GSE3141, GSE31908, GSE37745, and GSE50081. EGFR and sortilin expression data and corresponding overall patient survival and vital status data were extracted from the cBioPortal data set taken from TCGA archives (TCGA, Nature 2014; Provisional TCGA), representing data from a total of 522 lung adenocarcinoma patients.

**Statistical analysis**. Treatments, relative fluorescence intensities, antibody arrays, and western blotting experiments were compared with controls using the StatView software (v.5.0). Data shown are representative of at least three independent experiments. Error bars represent s.e.m. Results were analyzed for statistical significance by Student's t-test, log-rank (Mantel-Cox) T. $p \leq 0.05$ was considered significant, with actual values represented by asterisks (*$p \leq 0.05$; **$p \leq 0.01$; ***$p \leq 0.001$). Survival data were subjected to Kaplan−Meier analysis. EGFR and sortilin expression data were correlated using Cox regression analysis. Cutoff values were determined using the Cutoff Finder[66].

**Data availability**. All relevant data are available from the corresponding author on request.

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

## Acknowledgements

This study was generously supported by Chaire de Pneumologie Expérimentale from Association Limousine d'Aide aux Insuffisants Respiratoires-Assistance Ventilatoire à Domicile (ALAIR-AVD; Limoges, France), the Foundation of the University of Limoges, the Comité d'Orientation de la Recherche sur le Cancer en Limousin and the Ligue Contre le Cancer. H.A. was supported by a doctoral fellowship from ADER-LPC (Association du Développement Education Recherche-Limousin Poitou-Charentes). We are grateful to all of our colleagues have contributed their time and materials to this study. We thank Aurélie Lacroix and Nicolas Vedrenne for assistance with experiments on mice and human tissue samples. We are especially grateful to Alain Chaunavel for technical support from "Centre de Ressources Biologiques Biolim," Department of Pathology, University Hospital Limoges, and Claire Carrion from the Imaging Cytometry Platform of the University of Limoges.

## Author contributions

H.A.-A., T.N. performed the experiments and analyzed the data. A.M. contributed to image analysis. K.D., F.B. and B.M. participated in collection of patient's samples and clinical data collection. H.A.-A., T.N., F.V., F.B., and M.-O.J. participated in the study design. H.A.-A., T.N., F.V., F.L., and M.-O. J. coordinated the study. All authors read and approved the final manuscript.

## Additional information

**Competing interests:** The authors declare no competing financial interests.

