## [Peer Review File · Nature Communications]

Reviewers' comments:

Reviewer #1 (Remarks to the Author):

In this paper, the authors have investigated the fate of EGFR transfer from membrane to vesicles after EGF stimulation. They found that sortilin is critical in that its decreased expression, blocked EGFR at the plasma membrane and promoted cell growth. They also showed that loss of sortilin was correlated with poorer survival in lung adenocarcinoma.

The strength of the paper is that it is interesting and is relevant to a number of fields.

However there are a number of weaknesses that need to be addressed before the paper can be published.

Major comments

1/ This study is set on a defined time point (30min), however most of the signalings post EGF stimulation happened within 5 to 15 min maximum. Are the authors really analyzing the first interaction between sortilin and EGFR or a residual effect? Authors should provide a time course of EGFR/sortilin interaction at shorter times.

2/The domain of EGFR mediating the interaction with sortilin is not shown. This point is particularly important.

3/in this regard: Many point-mutations within the cytosolic domain of EGFR occur in lung cancer (ie L858R, T790M): are these mutations associated with a loss of interaction with sortilin? Authors should recapitulate their conclusions in other lung tumor cell-lines (including H1975, H1650) harboring EGFR mutations

4/ Authors claim that the EGFR stability at the plasma membrane sustains cell growth of sortilin-depleted cells. However, there is no evidence supporting this conclusion. EGFR mutants with deletion or mutations should be done to prove that in absence of interaction with sortilin, EGFR alone is able to sustain growth (Cf comment 2). Moreover, it will be important to show/discuss the "interactome" of sortilin in order to have an idea of the other receptors that may stay at the membrane when SORT1 is depleted and that could also explain the sustained cell growth.

Minor comments

1/ "reduction in EGFR levels following lysosomal degradation": Authors should show that the lysosomal degradation is indeed active in their model at the beginning of the study (not in suppl fig 3).

2/ Are the result similar in serum starved cells? It may change the conclusion of the study in a tumor environment depleted in abundance of growth factors.

3/P7 line 9-11: conclusion is complicated to understand, please rephrase.

4/ 2 isoforms of sortilin are expressed in cells, do all isoforms bind similarly EGFR?

5/ Authors used alternatively protein and mRNA for patient survival, however, they did not provide evidence that there is a correlation between mRNA and protein of EGFR and SORT1 in Lung Cancer patients.

6/ As the EGFR mutation status of patients are not presented, I will suggest to use the data from TCGA in discussion.

Reviewer #2 (Remarks to the Author):

This manuscript brings an important contribution to the understanding of EGFR deactivation after signal reception and transduction. This deactivation is known to occur through internalization and embedding into endosomes, but the detailed processes are incompletely deciphered. This paper shows how the membrane protein sortilin directly interacts with EGFR upon EGFR stimulation by its ligand. This takes place at the plasma membrane level before endosome embedding, as shown by using a sortilin construct devoid of its C-terminal part required for internalization. Sortilin depletion by shRNA constructs leads to the lack of EGFR internalization and increases EGFR signaling to the

MAP kinase pathway.

The consequences of this *in vitro* demonstration of the role of sortilin in EGFR signal transduction were clearly translated *in vivo* by xenografting tumor cells with sortilin knockdown in immunocompromised mice, which is accompanied by tumor growth acceleration; and in the clinics, first by immunohistochemical association between high proliferation markers and low sortilin expression, secondly by an *in silico* exploration of TCGA database showing an association between low sortilin expression and poor patient survival.

The demonstration provided by the authors on cultured cells with performant cell and molecular biology tools comes out very convincingly to the proposal of a new marker of lung cancer aggressiveness and prognosis, which emphasizes the importance of the work presented here.

The manuscript is well written and well balanced, each assertion being clearly supported by relevant experimentation. The only relatively weaker point is the absence of western blot verification of the knock down of sortilin in the experiments presented in Figure 4. Also, the increase of phosphorylation of AKT and ERK1/2 in control cells (Figure 4d) is not obvious on the western blots presented. It is at best transient for AKT as for ERK1/2.

Just to mention some minor points: two distinct paragraphs in the Materials and Methods section have the same title; reference to figure 3 instead of figure 2 and of IB instead of IP should be corrected at the end of the 3rd part of the Results section (the pages have not been numbered...)

Reviewer #3 (Remarks to the Author):

The present work investigated the role of sortilin in EGFR endocytosis and signaling. Sortilin was previously implicated in sorting pathways between PM, endosomes and TGN. In the present studies, the authors showed that sortilin interacts with EGFR by CoIP and that this interaction is increased upon EGF stimulation. The interaction - and its EGF dependency - was shown also by Duolink proximity ligation assay. In growing cells, sortilin and EGFR do not colocalize being EGFR mainly localized at the PM, while sortilin is primarily in the TGN. Upon EGF stimulation for 30 min, the EGFR relocalized in perinuclear area and partially colocalizes with sortilin and with lamp-positive compartment. The ablation of sortilin causes an increase of cell surface EGFR levels (and of total EGFR), and a delay in EGFR disappearance from the PM upon EGF. The authors conclude that sortilin promotes both constitutive as well as ligand-induced endocytosis of the EGFR and its trafficking towards lysosome, restricting EGFR signaling. Finally, they propose that sortilin levels might be relevant in lung adenocarcinomas with EGFR overexpression.

Despite some interesting - although preliminary - observations, the paper suffers of several weaknesses, both of conceptual and technical nature. The data provided do not support the main conclusions raised by the authors. For these reasons, the paper is not suitable for publication.

Major concerns:

1) EGFR and sortilin interact by coIP and duolink assay already in absence of EGF stimulation; however, they showed very little co-localization - if any - in cells not stimulated with EGF. EGF stimulation increases CoIP, duolink and colocalization between the two, suggesting that sortilin-EGFR complex is ligand induced. However, many conditions and controls are missing: i) no colocalization with endocytic markers is provided by comparing cells stimulated vs. cells not stimulated with EGF, to see whether EGF is inducing changes in sortilin localization compared to basal levels; ii) the triple colocalization of EGFR/sortilin and endocytic markers is provided only for Rab5; iii) only one time point of EGF stimulation is provided (30 min, which is quite a late time point considering that EGFR is activated and phosphorylated already at 2 min after EGF addition). Thus, it is not clear whether the interaction occurs at early phase of EGFR activation (i.e. at the PM), or at late phase.

Colocalizations are also poorly convincing since they are performed upon overexpression of sortilin, which seems to alter the endocytic compartment. Indeed both Rab5, Lamp2 and TGN46 showed a

weird localization in cells with sortilin-GFP overexpression compared to untransfected cells, with bigger and fewer perinuclear clusters (Fig. 1 e, g). These colocalization studies, again, are done upon EGF and no staining in absence of EGF is provided. From this set of data, it's very difficult to make any conclusion about sortilin localization.

Fractionation and optiprep gradient is also not conclusive (Fig. S1g), given that no separation is achieved and all markers are spread all along the gradient. For instance, Rab5 is not enriched in what they ascribed to early endosomal (EE) fractions. Furthermore, EGF treatment is not shifting the EGFR in the 'hypothetical' EE compartment.

2) To demonstrate that the interaction is occurring at the PM, authors used dynasore treatment to force the EGFR at the PM. This experiment has several problems:

- controls are missing, as for instance the control for the effectiveness of dynasore treatment in inhibiting EGFR internalization. EGFR internalization seems not to be blocked from the pictures provided. Bigger EGFR clusters at the cell periphery are observed (Fig. 2a), which might simply indicate defects in endosome trafficking/maturation and not a block in the internalization step (in Fig. 2c EGFR forms really aberrant clusters that seem to localize inside the cell). To show that dynasore is affecting EGFR internalization, authors could, for instance, stimulate cells with fluorescently-labeled EGF at different time points, +/- dynasore. Before fixation, an acid wash treatment could be performed to remove PM-bound EGF, to demonstrate that EGFR clusters at the cell periphery are endocytic structures stacked at the PM and still connected with it.

- Also sortilin seems to change localization upon dynasore treatment, although again no control are provided, e.g sortilin localization in absence of dynasore, +/- EGF... In addition only one cell is shown and analyzed (Fig. 2d), and conclusion are drawn on the basis of one inset of one cell.

In conclusion, it is not clear whether dynasore is forcing the localization and the interaction of EGFR/sortilin at the PM or in another compartment. The increased coIP, and the fact that it becomes independent on EGF, is far from having a physiological meaning. The conclusion that EGFR and sortilin interact at the PM, and that sortilin has a role in constitutive EGFR endocytosis cannot be reached on the basis of these experiments.

Similarly, sortilin mutants are really art factual. The deltaC mutant seems to be much more expressed than the WT construct, and it is still mainly localized in the TGN/ perinuclear area, with only minor localization at the PM.

3) It's not demonstrated whether sortilin has a direct role in EGFR internalization (or if it has a role in receptor sorting at the PM).

No real endocytic assay was performed to show that sortilin ablation is directly affecting EGFR internalization. Authors provided only a FACS where they infer effects on internalization indirectly, by measuring the EGFR levels at the PM. However, cell surface EGFR levels might be influenced by many factors beside internalization, including: EGFR sorting from the TGN (given also that sortilin was previously implicated in this pathway), rate of synthesis, recycling...

The authors showed that sortilin ablation increases the basal levels of both total and PM EGFR. The increased EGFR PM levels might cause saturation of the endocytic pathway (with delayed internalization), as well as ligand-independent activation of the EGFR and of its downstream signaling, even in absence of a direct role of sortilin in EGFR endocytosis.

4) Data on A549 injection in vivo are interesting as well as data on patients. However, the connection with EGFR overexpression is weak. Indeed, patients stratification on the basis of sortilin high vs. low expression in the cohort of patients with high EGFR expression do not worsen the survival curve compared to the total cohort of patients (compare Fig. 6 f vs. e). Survival curve in the cohort of patients with normal EGFR expression should be also analyzed to highlight possible differences.

Other points:

In Figure 1D colors are mislabeled (EGFR should be in green and sortilin in red)

In Material and methods, cell proliferation assay, instead of FACS assay, is indicated in the title.

Antibodies used for colocalization assays are not indicated (e.g. EGFR, Rab5, Lamp2...).

Reviewer #4 (Remarks to the Author):

NCOMMS-16-29233

Sortilin controls EGFR internalization and limits tumor growth

The authors study the role of sortilin (also called neurotensin (NT) receptor-3 (NTSR3)) in the signaling function, and intra cellular metabolism of EGFR in lung cancer cells. They perform a wide variety of cell biologic, biochemical, and trafficking studies to show that sortilin interacts with EGFR regulating its internalization, while loss of sortilin expression leads to enhanced EGFR pathway activity, signaling, cell proliferation and tumor growth cell growth in vitro and in vivo. In studies of 78 lung adenocarcinoma tumor specimens by immunohistochemistry they find that low sortilin expression is associated with more aggressive histology, and by use of deposited lung adenocarcinoma datasets they show that low sortilin expression in resected non-small cell lung cancers results in impaired survival in all tumors and in tumors with high EGFR expression. They conclude: "Thus, sortilin is a novel regulator of EGFR intracellular trafficking that acts by controlling receptor internalization and limiting tumor growth. "

Comments to the authors:

The experiments are all technically well done and I believe all of the trafficking and cell signaling studies. If everything they found is true, these studies could provide new insights into developing further EGFR targeted therapy for lung adenocarcinoma. There are several major issues the authors need to address.

1. Their presentation obviously suggests they believe sortilin is functioning as a tumor suppressor. From deposited data it would be useful to know if there are tumor mutations in sortilin and loss of heterozygosity, or epigenetic mediated loss of expression as seen for classical tumor suppressor genes. Whatever the answer is we need to know.
2. Of course there has been intense study of EGFR mutant lung adenocarcinomas. They present absolutely no data on correlations between the presence of EGFR mutations (and the often associated EGFR amplification) and expression of sortilin. Likewise, what about the relationship of sortilin expression to other common driver mutations in lung adenocarcinoma such as KRAS. Whatever the answer is we need to know.
3. The real importance of their observations comes from the cell biologic studies showing alterations in cell proliferation or tumor cell growth. The A549 cells have formed a tumor that presumably killed the patient it arose in with sortilin being expressed. If they are correct, they should be able to identify lung adenocarcinoma cell lines with low sortilin expression and by exogenously re-expressing sortilin demonstrate some kind of impairment of malignant behavior. Again, whatever the answer is we need to know.
4. While we currently use EGFR targeted therapy in EGFR mutant lung cancers, how would such therapy work in lung adenocarcinoma cell lines and xenografts that express high or low levels of sortilin? I could imagine various scenarios but to place their findings in context we need to know this type of data.
5. While their work was focused on lung adenocarcinoma it would be important to know if there are any differences in expression or survival in lung squamous cell cancers. Whatever the answer is we need to know.
6. In reviewing the literature it was interesting to see that in breast cancer, ovarian cancer and colon cancer preclinical models, exactly the opposite conclusions were reached – in that it appears that sortilin is acting like an oncogene and was considered as a therapeutic target. While one can argue about lineage differences in sortilin function, I found this difference amazing. Even more amazing was that this discrepancy was not discussed by the authors. This is particularly true given the similarities of EGFR driven lung cancer and Her2 driven breast cancers. (Roselli, S., et al

(2015) Sortilin is associated with breast cancer aggressiveness and contributes to tumor cell adhesion and invasion. *Oncotarget* 6, 10473-10486; Massa, F., et al. (2014) Impairment of HT29 Cancer Cells Cohesion by the Soluble Form of Neurotensin Receptor-3. *Genes & cancer* 5, 240-249; Ghaemimanesh, F., et al. (2014) The effect of sortilin silencing on ovarian carcinoma cells. *Avicenna J Med Biotechnol* 6, 169-177)

7. The label on their Figure 7 is given as "Figure 1."

We would like to thank the reviewers for their insightful comments on our study, which helped us significantly improve the manuscript. Detailed responses to the reviewers are provided below.

Point-by-point response to reviewers' comments

Reviewer #1 (Remarks to the Author):

In this paper, the authors have investigated the fate of EGFR transfer from membrane to vesicles after EGF stimulation. They found that sortilin is critical in that its decreased expression blocked EGFR at the plasma membrane and promoted cell growth. They also showed that loss of sortilin was correlated with poorer survival in lung adenocarcinoma. The strength of the paper is that it is interesting and is relevant to a number of fields. However there are a number of weaknesses that need to be addressed before the paper can be published.

Major comments

1/This study is set on a defined time point (30 min), however most of the signaling post EGF stimulation happened within 5 to 15 min maximum. Are the authors really analyzing the first interaction between sortilin and EGFR or a residual effect? Authors should provide a time course of EGFR/sortilin interaction at shorter times.

Accordingly, we performed PLA experiments following a time course of EGF stimulation (1, 2, 5, 15, and 30 min) in A549 cells. As expected, the PLA signal significantly increased after 2 min of EGF stimulation, and that increase was maintained for 30 min (Figure 1b and c). Those results provide additional evidence supporting the idea that sortilin interacts with EGFR at an early stage, when internalization is first triggered. For PLA experiments, we quantified the number of dots per cell instead of PLA signal intensity to standardize our results relative to the literature. The manuscript and figures have been updated (page 4, lines 92 and 97).

2/The domain of EGFR mediating the interaction with sortilin is not shown. This point is particularly important.

We showed that C-terminal truncated sortilin strongly interacts with EGFR in a HEK model (Figure 3e and f), suggesting that this interaction is mediated by the sortilin VPS10 domain, which constitutes its entire extracellular domain and has been implicated in several intracellular sorting processes (see Marcusson EG et al., Cell, 1994, PMID: 8187177; Cooper AA et al., J. Cell Biol., 1996, PMID: 8636229; Petersen CM et al., J. Biol. Chem., 1997, PMID: 9013611). To identify the EGFR domain mediating this interaction, we overexpressed both sortilin and the extra- or intracellular domain of EGFR in HEK cells. Interestingly, IP experiments showed that the EGFR extracellular domain interacted strongly with sortilin (Figure 3g). Therefore, we can conclude that the sortilin VPS10 extracellular domain interacts with the extracellular domain of EGFR. The manuscript has been updated to reflect the new results (page 7, lines 197-204).

3/in this regard: Many point-mutations within the cytosolic domain of EGFR occur in lung cancer (i.e. L858R, T790M): are these mutations associated with a loss of interaction with sortilin? Authors should recapitulate their conclusions in other lung tumor cell-lines (including H1975, H1650) harboring EGFR mutations

We agree that the investigation of EGFR–sortilin interaction in other NSCLC cell lines harboring point mutations or deletion in EGFR would provide greater confidence in the overall conclusion. Accordingly, we performed IP and PLA on lung cancer cell lines with EGFR point mutations (H1975,

H3255) or deletion (H1650). Our results revealed strong EGFR–sortilin interaction, even under basal conditions (Figure 3 h, i, and j). Indeed, mutations in the TK domain of EGFR mostly activated EGFR aberrantly and promoted its internalization, suggesting that the mutants' greater ability to be internalized might facilitate the EGFR–sortilin interaction (see Chung BM et al., *BMC Cell. Biol.*, 2009, PMID: 19948031; Tomas A et al., *Trends Cell. Biol.*, 2014, PMID: 24295852). Our results support a model in which the VPS10 domain of sortilin interacts with the EGFR-ECD, irrespective of somatic mutations in the EGFR tyrosine kinase domain. The manuscript has been updated to reflect the new results (page 7, lines 202-215).

4/ Authors claim that the EGFR stability at the plasma membrane sustains cell growth of sortilin-depleted cells. However, there is no evidence supporting this conclusion. EGFR mutants with deletion or mutations should be done to prove that in absence of interaction with sortilin, EGFR alone is able to sustain growth (Cf comment 2). Moreover, it will be important to show/discuss the “interactome” of sortilin in order to have an idea of the other receptors that may stay at the membrane when SORT1 is depleted and that could also explain the sustained cell growth.

We thank the reviewer for pointing out the necessity of confirming that EGFR alone sustains proliferation in sortilin-depleted A549 cells, as well as discussing the sortilin “interactome.” To investigate whether the high proliferation rate of sortilin-depleted cells was truly related to EGFR plasma membrane retention, we silenced EGFR expression using siRNA and followed EdU incorporation in these cells. Optimal EGFR silencing was reached 72 h after transfection. At that time, cells exhibited a reduced proliferation rate, similar to that of the control cells (Figure 4g), indicating that silencing of EGFR alone is sufficient to significantly decrease proliferation by sortilin-depleted cells. The manuscript has been updated to reflect the new results (page 8, lines 248-250).

As demonstrated previously, EGFR mutations did not prevent the EGFR–sortilin association, but instead promoted it, probably as a consequence of the more active EGFR internalization in the presence of the mutations, which mimic a constitutively active state (see Shan Y et al., *Cell*, 2012, PMID: 22579287). In regard to the comments of reviewers #1 and #4, and the analysis of deposited data, we showed that sortilin expression is tightly correlated with that of EGFR, and remains downregulated in the TKI-resistant cell line H1975, which harbors the double mutation L858R/T790M. These observations suggest that sortilin could act as a suppressor of EGFR signaling. Hence, with respect to this point (also raised by reviewer #4), we identified H1975 as a model cell line that is resistant to TKI and expresses a low level of endogenous sortilin (Figure 4j). Using these cells, we transiently overexpressed sortilin for 72 h and followed EdU incorporation. Interestingly, sortilin overexpression in H1975 cells decreased both AKT and ERK phosphorylation (Figure 4k), as well as cell proliferation (Figure 4l), thereby decreasing malignant cell behavior. The manuscript has been updated to reflect the new results (page 9, lines 260-270).

Consistent with the resistance of H1975 to gefitinib, we assessed proliferative signaling as reflected by MAP kinase activation following treatment with 1–10 μ M gefitinib. Strikingly, sortilin overexpression decreased the activation of the proliferative factor AKT, as evidenced by reduction of ERK phosphorylation in a dose-dependent manner (Figure 4m). In addition, we tried to stably overexpress sortilin in the same cell line, but found that this manipulation significantly affected cell proliferation. Indeed, sortilin knock-in drastically limited the proliferation of H1975 infected cells in comparison with control cells, preventing the establishment of stable sortilin knock-in cells, consistent with a crucial role for sortilin in suppressing EGFR signaling. Therefore, the sortilin interactome is not

relevant to our model. The manuscript has been updated to reflect the new results (page 9, lines 270-272).

Minor comments

1/ “reduction in EGFR levels following lysosomal degradation”: Authors should show that the lysosomal degradation is indeed active in their model at the beginning of the study (not in suppl fig 3).

We have revised the manuscript accordingly (Figure 4h, page 8, lines 253-257).

2/ Are the result similar in serum starved cells? It may change the conclusion of the study in a tumor environment depleted in abundance of growth factors.

As we understand the question, the reviewer is asking what happens to the EGFR–sortilin interaction in serum-starved cells. This is indeed an interesting question; however, under stress conditions, such as serum starvation, EGFR interacts strongly with lysosomal-associated transmembrane proteins that are mainly localized in early and late endosomes (see Meng Y et al., *Oncogene*, 2016, PMID: 27212036; Maki Y et al., *Sci. Rep.*, 2015, PMID: 26343532). Thus, EGFR is stabilized, resulting in endosomal arrest and receptor accumulation (Tan X et al., *EMBO. J.*, 2015, PMID: 25588945). Moreover, EGFR can inhibit autophagy, promoting its recycling (Wei Y et al., *Cell*, 2013, PMID: 24034250). Hence, all experiments were performed under normal serum conditions to avoid EGFR accumulation in non-degradative endosomes, and especially to avoid confusion between EGFR accumulation due to sortilin depletion and stress-induced EGFR accumulation. Nevertheless, we also performed PLA on serum-starved A549 cells, but our results revealed no significant alteration in the EGFR–sortilin interaction in comparison with control cells (data not shown).

3/P7 line 9-11: conclusion is complicated to understand, please rephrase.

We have carefully rewritten this section of the manuscript; we hope that the reviewer agrees that clarity has been achieved.

4/ 2 isoforms of sortilin are expressed in cells, do all isoforms bind similarly EGFR?

The Δ C-sortilin plasmid was engineered to block sortilin recycling from the plasma membrane to the intracellular organelles (Finan GM et al., *J. Biol. Chem.*, 2011, PMID: 21245145). Accordingly, Δ C-sortilin lacking the intracellular domain remains embedded in the plasma membrane. In the revised manuscript, we demonstrate that only the ECD of sortilin, containing the VPS10 domain, interacts with EGFR (Figure 3g). Together, our findings show that expression of Δ C-sortilin exhibits a stronger interaction with EGFR because the sortilin VPS10 domain remains artificially accumulated at the cell surface, despite the absence of EGF (Figure 3a, d, and e).

5/ Authors used alternatively protein and mRNA for patient survival, however, they did not provide evidence that there is a correlation between mRNA and protein of EGFR and SORT1 in Lung Cancer patients.

We completely agree with this point, and we are aware of that fact. Unfortunately, the accessibility of patient samples remains a constraint on clinical analyses. Specifically, accessibility is dependent on the individual case, as well as the size of the tumor fragment kept by the pathologists to establish the diagnosis.

6/ As the EGFR mutation status of patients are not presented, I will suggest to use the data from TCGA in discussion.

In response to this interesting comment, we analyzed the TCGA provisional data and extracted 33 of 524 patients with mutated EGFR. Of these, 24 harbored mutations in the tyrosine kinase domain: 3 with TKI-resistant mutations, and 21 with TKI-sensitive mutations or deletion. In the revised manuscript, we demonstrate that sortilin is overexpressed when EGFR is amplified (Figure 6 f and g), often corresponding to EGFR point mutations. Our statistical tests support the idea that sortilin is overexpressed in cells with EGFR TKI-sensitive mutations (Figure 6i). Consistent with these analyses, sortilin expression was higher in TKI-sensitive cell lines (H3255, H1650) than in the resistant cell line H1975. However, the number of patients in the TCGA database with TKI-resistant mutations is too low to establish statistical significance. The manuscript has been updated to reflect the new analyses (page 10 and 11, lines 319-337).

Reviewer #2 (Remarks to the Author):

This manuscript brings an important contribution to the understanding of EGFR deactivation after signal reception and transduction. This deactivation is known to occur through internalization and embedding into endosomes, but the detailed processes are incompletely deciphered. This paper shows how the membrane protein sortilin directly interacts with EGFR upon EGFR stimulation by its ligand. This takes place at the plasma membrane level before endosome embedding, as shown by using a sortilin construct devoid of its C-terminal part required for internalization. Sortilin depletion by shRNA constructs leads to the lack of EGFR internalization and increases EGFR signaling to the MAP kinase pathway. The consequences of this in vitro demonstration of the role of sortilin in EGFR signal transduction were clearly translated in vivo by xenografting tumor cells with sortilin knockdown in immunocompromised mice, which is accompanied by tumor growth acceleration; and in the clinics, first by immunohistochemical association between high proliferation markers and low sortilin expression, secondly by an in silico exploration of TCGA database showing an association between low sortilin expression and poor patient survival.

The demonstration provided by the authors on cultured cells with performant cell and molecular biology tools comes out very convincingly to the proposal of a new marker of lung cancer aggressiveness and prognosis, which emphasizes the importance of the work presented here.

The manuscript is well written and well balanced, each assertion being clearly supported by relevant experimentation. The only relatively weaker point is the absence of western blot verification of the knock down of sortilin in the experiments presented in Figure 4. Also, the increase of phosphorylation of AKT and ERK1/2 in control cells (Figure 4d) is not obvious on the western blots presented. It is at best transient for AKT as for ERK1/2.

We thank the reviewer for this suggestion, which we have carefully addressed in the revised manuscript. Accordingly, we have added western blot verification of both the knockdown of sortilin and the phosphorylation of AKT and ERK in control cells under basal conditions, in comparison with sortilin-depleted cells (Figure 4d). The manuscript has been updated to reflect the new results (page 8, lines 230-233).

Just to mention some minor points: two distinct paragraphs in the Materials and Methods section have the same title; reference to figure 3 instead of figure 2 and of IB instead of IP should be corrected at the end of the 3rd part of the Results section (the pages have not been numbered...)

We carefully rewrote these sections of the manuscript, and hope that the reviewer agrees that clarity has been achieved.

Reviewer #3 (Remarks to the Author):

The preset work investigated the role of sortilin in EGFR endocytosis and signaling. Sortilin was previously implicated in sorting pathways between PM, endosomes and TGN. In the present studies, the authors showed that sortilin interacts with EGFR by CoIP and that this interaction is increased upon EGF stimulation. The interaction - and its EGF dependency - was shown also by Duolink proximity ligation assay. In growing cells, sortilin and EGFR do not colocalize being EGFR mainly localized at the PM, while sortilin is primarily in the TGN. Upon EGF stimulation for 30 min, the EGFR relocated in perinuclear area and partially colocalizes with sortilin and with lamp-positive compartment. The ablation of sortilin causes an increase of cell surface EGFR levels (and of total EGFR), and a delay in EGFR disappearance from the PM upon EGF. The authors conclude that sortilin promotes both constitutive as well as ligand-induced endocytosis of the EGFR and its trafficking towards lysosome, restricting EGFR signaling. Finally, they propose that sortilin levels might be relevant in lung adenocarcinomas with EGFR overexpression. Despite some interesting - although preliminary - observations, the paper suffers of several weaknesses, both of conceptual and technical nature. The data provided do not support the main conclusions raised by the authors. For these reasons, the paper is not suitable for publication.

Major concerns:

1) EGFR and sortilin interact by coIP and duolink assay already in absence of EGF stimulation; however, they showed very little co-localization - if any - in cells not stimulated with EGF. EGF stimulation increases CoIP, duolink and colocalization between the two, suggesting that sortilin-EGFR complex is ligand induced. However, many conditions and controls are missing:

i) no colocalization with endocytic markers is provided by comparing cells stimulated vs. cells not stimulated with EGF, to see whether EGF is inducing changes in sortilin localization compared to basal levels;

We thank the reviewer for bringing this critical point to our attention. We agree that showing the Mander's coefficient for EGFR and sortilin colocalization with endocytic markers under the basal condition will help to clarify our conclusion for readers. In the revised manuscript, we show that, under basal conditions, sortilin is primarily localized in the TGN (Figure 1d, inset 6), in accordance with previously published results showing that less than 10% of total sortilin remains at the cell surface before being continually recycled back to the TGN through endosomes (Mazella J et al., Cell. Signal., 2001, PMID: 112574). On the other hand, in the absence of EGF stimulation, EGFR is mainly localized at the plasma membrane, whereas EGF stimulation elicits an increase in colocalization of sortilin and Rab5 (Figure 1d, insets 4 and 10; Figure 1e), indicating that a pool of sortilin is targeted toward Rab5+ early endosomes simultaneously with EGFR internalization, as attested by the increase of EGFR in early endosomes (Figure 1d, insets 1 and 7; Figure 1e). Thus, sortilin relocation upon EGF treatment implies that EGFR and sortilin colocalize in early endosomes during the triggering of EGFR endocytosis. Moreover, PLA assays performed over a time course of EGF stimulation revealed a significant EGFR-sortilin interaction starting 2 min after EGFR activation (Figure 1b and c), suggesting that the two proteins interact at an early stage of EGFR internalization. The manuscript has been updated to reflect the new results (page 4 and 5, lines 96 to 110).

ii) the triple colocalization of EGFR/sortilin and endocytic markers is provided only for Rab5;

We thank the reviewer for this comment, which helped us clarify the colocalization between EGFR–sortilin and the early endosome marker Rab5. Mander’s coefficient analyses (shown in Figure 1e) and the remarkable shift of sortilin toward early endosomes during EGF-induced EGFR internalization support the idea that the EGFR–sortilin interaction occurs in the early stage of endocytosis, excluding the possibility that the interaction is initiated in the lysosome. Thereafter, we sought to confirm that EGFR and sortilin colocalized together with Rab5. Because this study addresses the key role of sortilin in EGFR retrograde trafficking from the plasma membrane, we provided triple colocalization only with Rab5. This point has been carefully emphasized in the manuscript, and we hope that the reviewer would agree that we have clarified this point.

iii) only one time point of EGF stimulation is provided (30 min, which is quite a late time point considering that EGFR is activated and phosphorylated already at 2 min after EGF addition). Thus, it is not clear whether the interaction occurs at early phase of EGFR activation (i.e. at the PM), or at late phase.

We are very grateful to the reviewer for inviting us to clarify this important point. In the revised manuscript, we provide the results of PLA following a time course of EGFR stimulation (Figure 1b and c). As demonstrated by the PLA results, the EGFR–sortilin interaction occurs after 2 min of EGF stimulation, confirming that sortilin is involved in the early stage of EGFR endocytosis. The manuscript has been updated to reflect the new results (page 4, lines 92 and 97).

Colocalizations are also poorly convincing since they are performed upon overexpression of sortilin, which seems to alter the endocytic compartment. Indeed both Rab5, Lamp2 and TGN46 showed a weird localization in cells with sortilin-GFP overexpression compared to untransfected cells, with bigger and fewer perinuclear clusters (Fig. 1 e, g). These colocalization studies, again, are done upon EGF and no staining in absence of EGF is provided. From this set of data, it’s very difficult to make any conclusion about sortilin localization. Fractionation and OptiPrep gradient is also not conclusive (Fig. S1g), given that no separation is achieved and all markers are spread all along the gradient. For instance, Rab5 is not enriched in what they ascribed to early endosomal (EE) fractions. Furthermore, EGF treatment is not shifting the EGFR in the ‘hypothetical’ EE compartment.

In the revised manuscript, we show the colocalization of EGFR or sortilin with different organelle markers, as discussed above (in 1, i). Based on the increase in Mander’s coefficient for the colocalization of EGFR and sortilin with rab5, and the strong EGFR–sortilin interaction after 2 min of EGF stimulation, these results indicate that sortilin interacts with EGFR during the triggering of its retrograde trafficking.

2) To demonstrate that the interaction is occurring at the PM, authors used Dynasore treatment to force the EGFR at the PM. This experiment has several problems:

- controls are missing, as for instance the control for the effectiveness of Dynasore treatment in inhibiting EGFR internalization. EGFR internalization seems not to be blocked from the pictures provided. Bigger EGFR clusters at the cell periphery are observed (Fig. 2a), which might simply indicate defects in endosome trafficking/maturation and not a block in the

internalization step (in Fig. 2c EGFR forms really aberrant clusters that seem to localize inside the cell).

To show that Dynasore is affecting EGFR internalization, authors could, for instance, stimulate cells with fluorescently-labeled EGF at different time points, +/- Dynasore. Before fixation, an acid wash treatment could be performed to remove PM-bound EGF, to demonstrate that EGFR clusters at the cell periphery are endocytic structures stacked at the PM and still connected with it.

Also sortilin seems to change localization upon Dynasore treatment, although again no control are provided, e.g. sortilin localization in absence of Dynasore, +/- EGF... In addition only one cell is shown and analyzed (Fig. 2d), and conclusion are drawn on the basis of one inset of one cell.

In conclusion, it is not clear whether Dynasore is forcing the localization and the interaction of EGFR/sortilin at the PM or in another compartment. The increased coIP, and the fact that it becomes independent on EGF, is far from having a physiological meaning. The conclusion that EGFR and sortilin interact at the PM, and that sortilin has a role in constitutive EGFR endocytosis cannot be reached on the basis of these experiments. Similarly, sortilin mutants are really art factual. The deltaC mutant seems to be much more expressed than the WT construct, and it is still mainly localized in the TGN/ perinuclear area, with only minor localization at the PM.

We thank the reviewer for this comment. We believe that we have carefully described how the Dynasore behaves in EGFR endocytosis. Larger EGFR clusters are observed at the cell periphery because Dynasore acts between the formation of a deeply invaginated coated pit and membrane constriction to form a constricted coated pit that generates an early endosome following membrane fission (see Nankoe SR and Sever S, Trends Cell. Biol., 2006, PMID: 17064900). Thus, Dynasore elicits pre-accumulation of "U-Shaped" endocytic coated structures at the cell surface (see Macia E et al., Dev. Cell., 2006, PMID: 16740485; Newton AJ et al., PNAS, 2006, PMID: 17093045). In this manner, Dynasore inhibits EGFR endocytosis by inducing sustained EGFR signaling from the cell surface, as evidenced by EGFR hyperphosphorylation upon EGF stimulation, reflecting the disruption of EGFR degradation following complete internalization (Figure 2b). However, dynamins act on the intracellular traffic of several vesicles, such as the TGN, or on endosome maturation. Thus, upon inhibition of dynamins by Dynasore, the intracellular EGFR clusters may reflect the inhibition of maturation of already internalized endosomes, as shown by Figure 2a (insets 3-1, 3-2; colocalization with the EEA1 marker, Figure 1c), and the intracellular PLA signal (Figure 2e, insets 3-1, 3-2) (see Jaiswal JK et al., Cell, 2009, PMID: 19563761; Mesaki K et al., Plos One, 2011, PMID: 21572956). Furthermore, because sortilin cycles continually between the plasma membrane and the TGN through endosomes, inhibition of dynamins may interfere with its intracellular trafficking by blocking its recycling from the plasma membrane.

3) It's not demonstrated whether sortilin has a direct role in EGFR internalization (or if it has a role in receptor sorting at the PM).

No real endocytic assay was performed to show that sortilin ablation is directly affecting EGFR internalization. Authors provided only a FACS where they infer effects on internalization indirectly, by measuring the EGFR levels at the PM.

However, cell surface EGFR levels might be influenced by many factors beside internalization, including: EGFR sorting from the TGN (given also that sortilin was previously implicated in this pathway), rate of synthesis, recycling...

The authors showed that sortilin ablation increases the basal levels of both total and PM EGFR. The increased EGFR PM levels might cause saturation of the endocytic pathway (with delayed internalization), as well as ligand-independent activation of the EGFR and of its downstream signaling, even in absence of a direct role of sortilin in EGFR endocytosis.

We thank the reviewer for pointing out the necessity of confirming whether sortilin plays a direct role in EGFR internalization or functions as a sorting receptor. Accordingly, we transiently transfected Δ C-sortilin lacking the intracellular domain, which contains a peptide signal recognized by retromer proteins; thus sortilin trafficking is dependent on its cytoplasmic tail. Indeed, Δ C-sortilin was engineered to impair the sortilin membrane-sorting function, and consequently remains embedded at the plasma membrane (see Finan GM et al., J. Biol. Chem., 2011, PMID: 21245145; Evans SF et al., J. Biol. Chem., 2011, PMID: 21730062; Wilson et al., J. Cell. Sci., 2014, PMID: 25037567). As demonstrated, Δ C-sortilin-GFP overexpression exhibits plasma membrane staining. Interestingly, Δ C-sortilin interacts strongly with EGFR even in the absence of EGF. IP experiments following overexpression of both sortilin and the extra- or intracellular domain of EGFR revealed that the EGFR extracellular domain interacts with sortilin.

Moreover, in sortilin-depleted A549 cells, EGFR remains at the plasma membrane, excluding the possibility that sortilin depletion impairs EGFR plasma membrane anterograde transport following post-translational modifications in the TGN. To further rule out the possibility that the EGFR–sortilin interaction occurs after post-translational modifications, we treated A549 cells with the protein synthesis inhibitor cycloheximide (CHX). No differences in the EGFR–sortilin interaction were observed following EGF stimulation in the presence or absence of CHX, arguing that the interaction occurs in the early stage of EGFR internalization and not as a result of post-translational modifications. Consistent with these results, and because sortilin interacts with EGFR through its VPS10 domain, we performed FACS analysis to monitor cell surface EGFR internalization, as described by several authors (see Garvalov BK et al., Nat. Commun., 2014, PMID: 25420589; Ning L et al., Methods Mol. Biol., 2008, PMID: 19066037).

As shown previously, despite the function of sortilin in the TGN pathway, the EGFR–sortilin interaction was independent of both EGFR and sortilin synthesis

4) Data on A549 injection in vivo are interesting as well as data on patients. However, the connection with EGFR overexpression is weak. Indeed, patients stratification on the basis of sortilin high vs. low expression in the cohort of patients with high EGFR expression do not worsen the survival curve compared to the total cohort of patients (compare Fig. 6 f vs. e). Survival curve in the cohort of patients with normal EGFR expression should be also analyzed to highlight possible differences.

In regard to this interesting point, which was also raised by reviewer #4, we analyzed the TCGA provisional data to investigate whether sortilin expression depends on oncogenic drivers such as KRAS or EGFR amplification, as observed in lung adenocarcinoma. Although our statistical tests revealed no significance of KRAS mutations, they emphasize that sortilin is overexpressed when EGFR is highly expressed. Based on these analyses, we performed quantitative PCR on a cohort of

patients with low or high EGFR expression levels. Our analysis supported the TCGA data, confirming that sortilin expression is significantly higher in patients with EGFR amplification; we have added these results to Figure 6. A large majority (75%) of cases with increased gene copy number have *EGFR* mutations (see Ladanyi M and Pao W, *Mol. Pathol.*, 2008, PMID: 18437168; Li AR et al., *J. Mol. Diagn.*, 2008, PMID: 18403609). Lung adenocarcinomas with mutated *EGFR* exhibit significant responses to EGFR receptor tyrosine kinase inhibitors (TKIs) such as gefitinib (see Lynch TJ et al., *N. Engl. J. Med.*, 2004, PMID: 15118073; Paez JG et al., *Science*, 2004, PMID: 15118125; Pao W et al., *PNAS*, 2004, PMID: 15329413). However, in cancers treated with TKIs, acquired resistance mutations occur, most commonly the threonine-to-methionine shift at codon 790 (T790M). This type of mutation has been reported in tumors exhibiting acquired resistance to gefitinib, which is associated with poor patient outcome (see Lynch TJ et al., *N. Engl. J. Med.*, 2004, PMID: 15118073; Balak MN et al., *Clin. Cancer Res.*, 2006, PMID: 17085664). We showed that sortilin is expressed at higher levels in two TKI-sensitive EGFR-mutated NSCLC cell lines than in the gefitinib-resistant cell line H1975. Interestingly, overexpression of sortilin sensitizes H1975 to gefitinib (Figure 4m).

Furthermore, in the large cohort of TCGA data, several patients exhibited an increase in sortilin expression with no EGFR amplification, suggesting that sortilin might act on other tyrosine kinase receptors frequently upregulated in lung adenocarcinoma. However, this potential expansion of the influence of sortilin on all pro-oncogenic tyrosine kinase receptors remains speculative, and must be tested in future collaborative studies. Furthermore, we observed that sortilin undergoes genetic alterations, such as loss of heterozygosity, and the frequency of these events increased significantly between stages I and III ($p=0.0010$), supporting the idea that sortilin expression is reduced in advanced cancers (data not shown).

Other points: In Figure 1D colors are mislabeled (EGFR should be in green and sortilin in red) In Material and methods, cell proliferation assay, instead of FACS assay, is indicated in the title.

Antibodies used for colocalization assays are not indicated (e.g. EGFR, Rab5, Lamp2...).

We regret these typographical errors in the previous version of the manuscript, and thank the reviewer for these comments. We hope that the reviewer agrees that the writing in the revised manuscript has improved.

Reviewer #4 (Remarks to the Author):

NCOMMS-16-29233

Sortilin controls EGFR internalization and limits tumor growth

The authors study the role of sortilin (also called neurotensin (NT) receptor-3 (NTSR3)) in the signaling function, and intra cellular metabolism of EGFR in lung cancer cells. They perform a wide variety of cell biologic, biochemical, and trafficking studies to show that sortilin interacts with EGFR regulating its internalization, while loss of sortilin expression leads to enhanced EGFR pathway activity, signaling, cell proliferation and tumor growth cell growth in vitro and in vivo. In studies of 78 lung adenocarcinoma tumor specimens by immunohistochemistry they find that low sortilin expression is associated with more aggressive histology, and by use of deposited lung adenocarcinoma datasets they show that low sortilin expression in resected non-small cell lung cancers results in impaired survival in all tumors and in tumors with high EGFR expression. They conclude: “Thus, sortilin is a novel regulator of EGFR intracellular trafficking that acts by controlling receptor internalization and limiting tumor growth. “

Comments to the authors: The experiments are all technically well done and I believe all of the trafficking and cell signaling studies. If everything they found is true, these studies could provide new insights into developing further EGFR targeted therapy for lung adenocarcinoma. There are several major issues the authors need to address.

1. Their presentation obviously suggests they believe sortilin is functioning as a tumor suppressor. From deposited data it would be useful to know if there are tumor mutations in sortilin and loss of heterozygosity, or epigenetic mediated loss of expression as seen for classical tumor suppressor genes. Whatever the answer is we need to know.

We thank the reviewer for the insightful suggestion that sortilin might be a tumor suppressor. Accordingly, we have analyzed publicly available data from the TCGA website for putative loss of heterozygosity (LOH), mutations, or methylation to provide biological explanations for the loss of sortilin expression in advanced lung adenocarcinoma, e.g., between early stage (I) and stage III. The sortilin gene (*SORT1*) is located on chromosome 1 at locus 1p13.3, flanked by the *MYBPHL* (Myosin Binding Protein H Like) and *PSMA5* (Proteasome Subunit Alpha 5) genes. After validating that all patients exhibited the same putative LOH for sortilin and its neighboring genes, our statistical analysis revealed that LOH occurs since the early stage I, and that its frequency is significantly elevated by stage III ($p=0.0010$ between stage I and III). *SORT1* mutations occur in 0.4% of 19,240 samples analyzed in COSMIC v62, and only one patient was detected among 524 cases in the TCGA database, suggesting that mutations did not represent a causal factor that would explain the reduced expression or loss of function of *SORT1*. Likewise, we detected no significant difference in *SORT1* promoter methylation between stages I and III. However, data about *SORT1* promoter methylation in control tissue from the same patient are not provided by TCGA.

2. Of course there has been intense study of EGFR mutant lung adenocarcinomas. They present absolutely no data on correlations between the presence of EGFR mutations (and the often associated EGFR amplification) and expression of sortilin. Likewise, what about the relationship of sortilin expression to other common driver mutations in lung adenocarcinoma such as KRAS. Whatever the answer is we need to know.

We thank the reviewer for these valuable suggestions. We have analyzed the deposited data to look for a possible correlation between sortilin expression and common oncogenic drivers in lung adenocarcinoma. Using data from TCGA, we found no correlation between sortilin expression and the KRAS mutations. Interestingly, we found that sortilin is significantly highly expressed in patients with amplified EGFR (Figure 6f), which has been confirmed in a cohort of patients with or without amplified EGFR (Figure 6g).

3. The real importance of their observations comes from the cell biologic studies showing alterations in cell proliferation or tumor cell growth. The A549 cells have formed a tumor that presumably killed the patient it arose in with sortilin being expressed. If they are correct, they should be able to identify lung adenocarcinoma cell lines with low sortilin expression and by exogenously re-expressing sortilin demonstrate some kind of impairment of malignant behavior. Again, whatever the answer is we need to know.

We thank the reviewer for this interesting comment. In regard to this comment and our analysis of the deposited data, we identified that sortilin is tightly correlated with EGFR expression and remains downregulated in the TKI-resistant cell line H1975 harboring the double mutation L858R/T790M. These observations suggest that sortilin could act as a suppressor of EGFR signaling. Hence, we identified H1975 as a model cell line that is resistant to TKI and expresses a low level of endogenous sortilin (Figure 4j). In these cells, we transiently overexpressed sortilin for 72 h and followed EdU incorporation. Interestingly, sortilin overexpression in H1975 cells decreased both AKT and ERK phosphorylation (Figure 4k), as well as cell proliferation (Figure 4l), thereby decreasing malignant cell behavior.

Consistent with their resistance to gefitinib, we assessed proliferative signaling as reflected by MAP kinase activation following treatment with 1–10 μ M gefitinib. Strikingly, sortilin overexpression decreased the activation of the proliferative factor AKT, as evidenced by reduction of ERK phosphorylation in a dose-dependent manner (Figure 4m). In addition, we tried to stably overexpress sortilin in the same cell line, but found that this manipulation significantly affected cell proliferation. Indeed, sortilin knock-in drastically limited the proliferation of H1975-infected cells in comparison with control cells, preventing the establishment of stable sortilin knock-in cells, consistent with a crucial role for sortilin in suppressing EGFR signaling. The manuscript has been updated to reflect the new results (page 9, lines 263-274).

4. While we currently use EGFR targeted therapy in EGFR mutant lung cancers, how would such therapy work in lung adenocarcinoma cell lines and xenografts that express high or low levels of sortilin? I could imagine various scenarios but to place their findings in context we need to know this type of data.

In vitro experiments were consistent with a role for sortilin in the response to gefitinib. We show in two TKI-sensitive EGFR-mutated NSCLC cell lines that sortilin remains highly expressed in comparison with the Gefitinib-resistant cell line H1975 (Figure 4j).

In H1975, which expresses low levels of sortilin expression and is resistant to gefitinib, sortilin overexpression inhibited proliferation and EGFR phosphorylation (Figure 4l and m, without gefitinib exposure). Strikingly, sortilin overexpression sensitized H1975 cells to gefitinib, as confirmed by the decrease in proliferative signaling (reflected by MAP kinase activation) following treatment with 1–10 μ M gefitinib. In parallel, the gefitinib response was demonstrated by a reduction in ERK phosphorylation in a dose-dependent manner (Figure 4m). In addition, we tried to stably overexpress sortilin in the same cell line, but found that it significantly affected cell proliferation. Indeed, sortilin knock-in drastically limited the proliferation of H1975-infected cells in comparison with control cells, preventing the establishment of stable sortilin knock-in cells, consistent with a crucial role for sortilin in suppressing EGFR signaling. Consequently, we are not able to obtain a sufficient amount of cells to achieve xenografts.

5. While their work was focused on lung adenocarcinoma it would be important to know if there are any differences in expression or survival in lung squamous cell cancers. Whatever the answer is we need to know.

We thank the reviewer for this useful comment. Using data from TCGA, we compared the level of sortilin mRNA expression between adenocarcinoma and squamous carcinoma. No significant variation in sortilin expression was observed between the different stages of squamous carcinoma, nor did sortilin expression confer any relative benefit on overall survival.

6. In reviewing the literature it was interesting to see that in breast cancer, ovarian cancer and colon cancer preclinical models, exactly the opposite conclusions were reached – in that it appears that sortilin is acting like an oncogene and was considered as a therapeutic target. While one can argue about lineage differences in sortilin function, I found this difference amazing. Even more amazing was that this discrepancy was not discussed by the authors. This is particularly true given the similarities of EGFR driven lung cancer and Her2 driven breast cancers. (Roselli, S., et al (2015) Sortilin is associated with breast cancer aggressiveness and contributes to tumor cell adhesion and invasion. *Oncotarget* 6, 10473-10486; Massa, F., et al. (2014) Impairment of HT29 Cancer Cells Cohesion by the Soluble Form of Neurotensin Receptor-3. *Genes & cancer* 5, 240-249; Ghaemimanesh, F., et al. (2014) The effect of sortilin silencing on ovarian carcinoma cells. *Avicenna J Med Biotechnol* 6, 169-177).

Accordingly, we agree with reviewer comments in breast and ovarian cancers, sortilin overexpression is correlated with poor prognosis (Roselli S et al., *Oncotarget*, 2015, PMID: 25871389; Ghaemimanesh, *Avicenna J Med Biotechnol*, 2014, PMID: 26683178). However, sortilin displays opposite functions depending on its secretion and cell cancer subtypes. Indeed, in colorectal cancer cells, the form of sortilin involved in cell detachment and invasion is a soluble form of sortilin released into the cell culture medium which could induce metastasis, by contrast with our results (Massa et al., *Genes Cancer*, 2014, PMID 27834811; Beraud-Dufour et al., *Int J Mol Sci*, 2016, PMID: 27834811;). In prostate cancer cells, the loss of sortilin expression may contribute to prostate cancer progression

(Tanimoto et al., Endocrinology, 2015, PMID: 25365768) suggesting that the function of sortilin in cancer is variable and could depend on cancer cell subtypes. Therefore, depending on the cellular model, sortilin might undergo post-translational modifications that affect its structure, which is tightly linked to its function.

7. The label on their Figure 7 is given as “Figure 1.”

We carefully renumbered the figures in the revised manuscript, and hope that the reviewer agrees that clarity has been achieved.

Reviewers' Comments:

Reviewer #1:

Remarks to the Author:

All comments raised has been answered in this new version. The article from Al Akhrass, Naves et al has been greatly improved. I recommend this article for publication.

Two minors points:

* I would like a line to discuss the results dealing with Fig 3i and Fig 4k-m with H1975 cell line to give a possible interpretation of the results.

* I would like authors to give the exact reference (ref or clone) of each antibodies used in the study. It is not clear for some providers witch antibodies they have used

Reviewer #2:

Remarks to the Author:

This manuscript has been considerably improved since the original submission: new results aiming at answering all the questions raised by the reviewers, careful corrections of the minor defects, rewriting of some awkward sentences.

I my opinion, this paper is now suitable for publication in Nature Communications.

Reviewer #3:

Remarks to the Author:

In the revised version of the manuscript, authors have performed additional experiments that have improved their work and help to clarify some points.

I have only few remaining issues:

I still believe that, to justify the title of the manuscript, a direct EGFR internalization assay, by following florescent or radioactive ligand at different time points upon sortilin KD, will corroborate the role of sortilin in the early steps of EGFR internalization. Extrapolation from what remain on the surface is a good indication (as in Fig. 4C), but can be affected by the initial EGFR level at the PM.

Authors also did not comment to the fact that they have performed all colocalizations upon overexpression of sortilin-GFP, which seems to alter significantly the endocytic compartments, such as rab5- and lamp2-positive compartments, greatly enlarged upon sortilin overexpression. This could be a relevant effect or an artifact due to overexpression. Colocalizations in GFP-sortilin low-expressing clones should clarify whether sortilin physiologically relocalizes to Rab5 compartment (if following the endogenous protein is unfeasible).

Finally, the dynasore experiment is not convincing to me. Indeed, as the authors also claim in their reply, dynasore acts at different steps and it is not only affecting internalization from the PM (but also endosome maturation, TGN trafficking...). Indeed, from a simple immunofluorescence it cannot be distinguished if the structures that accumulate in dynasore-treated cells are U-shaped structures connected to the PM or other trafficking intermediates. Thus, dynasore treatment cannot be used in PLA experiments to demonstrate that the interaction between EGFR and sortilin occurs at the PM. Given that authors have now provided evidences for the EGFR-sortilin interaction to occur already at 2 min of EGF stimulation (when the EGFR is still mainly at the PM), I would suggest either to remove this experiment or to critically discuss it in the text to explain its possible caveats.

Reviewer #4:

Remarks to the Author:

The authors have responded appropriately and in detail to all of the four reviewers' comments including presenting significant additional data. However, if I were the authors I would have put in one or two sentences in the discussion indicating that "sortillin in the mutant EGFR lung cancer context appears to be acting as a tumor suppressor or inhibits malignant behavior, while in other cancers it appears to act as an oncogene or promote malignant behavior. These differences will need to be resolved in future studies."

We would like to thank the reviewers for their insightful comments on our study, which helped us significantly improve the manuscript. Detailed responses to the reviewers are provided below.

Point-by-point response to reviewers' comments

Reviewer #1

All comments raised have been answered in this new version. The article from Al Akhrass, Naves et al has been greatly improved. I recommend this article for publication.

Two minors points:

*** I would like a line to discuss the results dealing with Fig 3i and Fig 4k-m with H1975 cell line to give a possible interpretation of the results.**

We carefully added a sentence concerning the likely explanation for these results which is underlined in yellow in the new version of our manuscript (lines 274 to 276).

*** I would like authors to give the exact reference (ref or clone) of each antibodies used in the study. It is not clear for some providers witch antibodies they have used**

References for antibodies used in this study have been carefully added in this version and highlighted in yellow, Page 13; section "Immunoblotting and Immunoprecipitation", Page 14; section "Cell proliferation assays and flow cytometry", Page 15; section "Immunofluorescence, PLA and confocal microscopy analysis" and Page 16; section "Patients and immunohistochemistry".

Reviewer #2 (Remarks to the Author):

This manuscript has been considerably improved since the original submission: new results aiming at answering all the questions raised by the reviewers, careful corrections of the minor defects, rewriting of some awkward sentences.

In my opinion, this paper is now suitable for publication in Nature Communications.

Reviewer #3 (Remarks to the Author):

In the revised version of the manuscript, authors have performed additional experiments that have improved their work and help to clarify some points.

I have only few remaining issues:

I still believe that, to justify the title of the manuscript, a direct EGFR internalization assay, by following florescent or radioactive ligand at different time points upon sortilin KD, will corroborate the role of sortilin in the early steps of EGFR internalization. Extrapolation from what remain on the surface is a good indication (as in Fig. 4C), but can be affected by the initial EGFR level at the PM.

In the present manuscript, we showed that sortilin-EGFR interaction is significantly increased since 2 min following EGF stimulation. FACS experiments performed indicate that EGFR remains on the cell surface following EGF stimulation. However, we agree that following fluorescent ligand is a direct approach within this context, hence we performed internalization assay by using EGF complexed to

Alexa Fluor 647 and monitored its internalization at 5 and 15 minutes both in control and sortilin-depleted cells. Our results supported that sortilin depletion induces a delay in EGFR internalization upon EGF stimulation in comparison with control cells indicating that sortilin acts in the early stage of EGFR internalization. We carefully added these results in supplementary data (supplementary Figure S2a) and added a sentence in the main text (lines 231-233) underlined in yellow.

Authors also did not comment to the fact that they have performed all colocalizations upon overexpression of sortilin-GFP, which seems to alter significantly the endocytic compartments, such as rab5- and lamp2-positive compartments, greatly enlarged upon sortilin overexpression. This could be a relevant effect or an artifact due to overexpression. Colocalizations in GFP-sortilin low-expressing clones should clarify whether sortilin physiologically relocates to Rab5 compartment (if following the endogenous protein is unfeasible).

We regret having missed to reply to this comment in the previous point-by-point letter. In the previous manuscript we explained why we overexpress sortilin; lines 109 to 112 “However, to determine the subcellular localization of sortilin, we transfected A549 cells with a sortilin-GFP fusion protein, thereby avoiding cross-reactivity with the secondary antibody”.

Indeed, sortilin-GFP overexpression did not affect the sortilin intracellular distribution as shown by Finan et al., who generously provided us the plasmid (Finan GM et al, J. Biol. Chem., 2011, PMID: 21245145) as mentioned in our materials and methods. Moreover, sortilin-GFP follows an intracellular itinerary that is typically related to this protein, as described in the published literature (Nykjaer A and, Willnow TE, Trends Neurosci., 2012, PMID: 22341525; Beraud-Dufour et al., Int J Mol Sci, 2016, PMID: 27834811). Furthermore, the magnification used for sortilin-GFP cells remains different, that affects the size of the intracellular compartments but increases the reader’s appreciation for sortilin distribution.

Finally, the dynasore experiment is not convincing to me. Indeed, as the authors also claim in their reply, dynasore acts at different steps and it is not only affecting internalization from the PM (but also endosome maturation, TGN trafficking...). Indeed, from a simple immunofluorescence it cannot be distinguished if the structures that accumulate in dynasore-treated cells are U-shaped structures connected to the PM or other trafficking intermediates. Thus, dynasore treatment cannot be used in PLA experiments to demonstrate that the interaction between EGFR and sortilin occurs at the PM. Given that authors have now provided evidences for the EGFR-sortilin interaction to occur already at 2 min of EGF stimulation (when the EGFR is still mainly at the PM), I would suggest either to remove this experiment or to critically discuss it in the text to explain its possible caveats.

We decided to keep Dynasore experiments in the main text with critics concerning the possible effect of the Dynasore on the other trafficking intermediates that could induce caveats close to the plasma membrane (lines 163-167).

Reviewer #4 (Remarks to the Author):

The authors have responded appropriately and in detail to all of the four reviewers' comments including presenting significant additional data. However, if I were the authors I would have put in one or two sentences in the discussion indicating that "sortillin in the mutant EGFR lung cancer context appears to be acting as a tumor suppressor or inhibits malignant behavior, while in other cancers it appears to act as an oncogene or promote malignant behavior. These differences will need to be resolved in future studies."

We thank the reviewer for this interesting comment that we carefully added to the discussion of this version (Line 380 to 383 underlined in yellow). We added references highlighting the role of sortilin as oncogene in other cancers. (Roselli S et al., Oncotarget, 2015, PMID: 25871389; Ghaemimanesh, Avicenna J Med Biotechnol, 2014, PMID: 26683178, Massa et al., Genes Cancer, 2014, PMID: 25221642 and Beraud-Dufour et al., Int J Mol Sci, 2016, PMID: 27834811).

Reviewers' Comments:

Reviewer #3:

Remarks to the Author:

The authors have addressed all my concerns. I think that the manuscript is suitable for publication on Nature Communication.